# Causal decomposition in the mutual causation system

Albert C. Yang [1,2,3], Chung-Kang Peng [1] & Norden E. Huang[4,5]

Inference of causality in time series has been principally based on the prediction paradigm. Nonetheless, the predictive causality approach may underestimate the simultaneous and reciprocal nature of causal interactions observed in real-world phenomena. Here, we present a causal-decomposition approach that is not based on prediction, but based on the covariation of cause and effect: cause is that which put, the effect follows; and removed, the effect is removed. Using empirical mode decomposition, we show that causal interaction is encoded in instantaneous phase dependency at a specific time scale, and this phase dependency is diminished when the causal-related intrinsic component is removed from the effect. Furthermore, we demonstrate the generic applicability of our method to both stochastic and deterministic systems, and show the consistency of causal-decomposition method compared to existing methods, and finally uncover the key mode of causal interactions in both modelled and actual predator–prey systems.

[1] Division of Interdisciplinary Medicine and Biotechnology, Beth Israel Deaconess Medical Center/Harvard Medical School, Boston, MA 02215, USA. [2] Institute of Brain Science, National Yang-Ming University, 11221 Taipei, Taiwan. [3] Department of Psychiatry, Taipei Veterans General Hospital, 11217 Taipei, Taiwan. [4] Center for Dynamical Biomarkers and Translational Medicine, National Central University, 32001 Chungli, Taiwan. [5] Key Laboratory of Data Analysis and Applications, First Institute of Oceanography, SOA, 266061 Qingdao, China. Correspondence and requests for materials should be addressed to A.C.Y. (email: cyang1@bidmc.harvard.edu)

Since the philosophical inception of causality by Galilei[1] and Hume[2] that cause must precede the effect in time, the scientific criteria for assessing causal relationships between two time series have been dominated by the notion of prediction, as proposed by Granger[3]. Namely, the causal relationship from variable $A$ to variable $B$ is inferred if the history of variable $A$ is helpful in predicting the value of variable $B$, rather than using information from the history of variable $B$ alone.

Granger causality is based on the time dependency between cause and effect[4]. As discussed by Sugihara et al.[5], Granger causality is critically dependent on the assumption that cause and effect are separable[3]. While the separability is often satisfied in linear stochastic systems where Granger causality works well, it might not be applicable in nonlinear deterministic systems where separability appears to be impossible because both cause and effect are embedded in a non-separable higher dimension trajectory[6,7]. Consequently, Sugihara et al.[5] proposed the convergent cross-mapping (CCM) method based on state-space reconstruction. In this context, cause and effect are state dependent, and variable $A$ is said to causally influence variable $B$, although counterintuitive, if the state of variable $B$ can be used to predict the state of variable $A$ in the embedded space, and this predictability improves (i.e., converges) as the time series length increases.

Existing methods of detecting causality in time series are predominantly based on the Bayesian[8] concept of prediction. However, cause and effect are likely simultaneous[9]. The succession in time of the cause and effect is produced because the cause cannot achieve the totality of its effect in one moment. At the moment when the effect first manifests, it is always simultaneous with its cause. Moreover, most real-world causal interactions are reciprocal; examples include predator–prey relationships and the physiologic regulation of body functions. In this sense, predictive causality may fail because the attempt to estimate the effect with the history of cause is compromised as the history of the cause is already simultaneously influenced by the effect itself, and vice versa.

Another constraint of the generalised prediction framework is that it requires a priori knowledge of the extent of past history that may influence and predict the future, such as the time lag between cause and effect in Granger's paradigm, or the embedding dimensions in state-space reconstructions such as CCM. Furthermore, a causality assessment is incomplete if it is based exclusively on time dependency or state dependency. Time series commonly observed in nature, including those from physiologic system or spontaneous brain activity, contain oscillatory components within specific frequency bands[10,11]. Identification of frequency-specific causal interaction is essential to understand the underlying mechanism[12,13]. Furthermore, the application of either linear Granger causality or the nonlinear CCM method alone is insufficient to accommodate the complex causal compositions typically observed in real-world data blending with oscillatory stochastic and deterministic mechanisms.

Here, we present a causal-decomposition analysis that is not based on prediction, and more importantly, is neither based on time dependency nor state dependency, but based on the instantaneous phase dependency between cause and effect. The causal decomposition essentially involves two assumptions: (1) any cause–effect relationship can be quantified with instantaneous phase dependency between the source and target decomposed as intrinsic components at specific time scale, and (2) the phase dynamics in the target originating from the source are separable from the target itself. We define the cause–effect relationship between two time series according to the covariation principle of cause and effect[1]: cause is that which put, the effect follows; and removed, the effect is removed; thus, variable $A$

causes variable $B$ if the instantaneous phase dependency between $A$ and $B$ is diminished when the intrinsic component in $B$ that is causally related to $A$ is removed from $B$ itself, but not vice versa. To achieve this, we use the ensemble empirical mode decomposition (ensemble EMD)[14–16] to decompose a time series into a finite number of intrinsic mode functions (IMFs) and identify the causal interaction that is encoded in instantaneous phase dependency between two time series at a specific time scale. We validate the causal-decomposition method with both stochastic and deterministic systems and illustrate its application to ecological time series data of prey and predators.

## Results

**Illustration of the causal-decomposition method.** Figure 1 depicts how the causal decomposition can be used to identify the predator–prey causal relationship of *Didinium* and *Paramecium*[17]. Briefly, we decomposed the time series of *Didinium* and *Paramecium* into two set of IMFs, and determined the instantaneous phase coherence[18] between comparable IMFs from the two time series (Fig. 1a). Orthogonality and separability tests were performed to determine the ensemble EMD parameter (i.e., added noise level) that minimises the nonorthogonal leakage and root-mean-square of the correlation between the IMFs, thereby ensuring the orthogonality and separability of the IMFs (Fig. 1d, e). Subsequently, we removed one of the IMFs (e.g., IMF 2) from *Paramecium* (Fig. 1b; subtract IMF 2 from the original *Paramecium* signal) and redecomposed the time series. We then calculated the phase coherence between the original IMFs of *Didinium* and redecomposed IMFs of *Paramecium*. This decomposition and redecomposition procedure was repeated for IMF 2 of *Didinium* (Fig. 1c) and generalised to all IMF pairs. This procedure enabled us to examine the differential effect of removing a causal-related IMF on the redistribution of phase dynamics in cause-and-effect variables. The relative ratio of variance-weighted Euclidian distance between the phase coherence of the original IMFs (i.e., Fig. 1a) and redecomposed IMFs (i.e., Fig. 1b, c) is therefore an indicator of causal strength (Fig. 1f), where a ratio of 0.5 indicates either no causality is detected or no difference in causal strength in the case of reciprocal causation, and a ratio approaching 0 or 1 indicates a strong causal influence from either variable $A$ or variable $B$, respectively.

**Application to deterministic and stochastic models.** Figure 2 depicts the causal-decomposition analysis in both deterministic[5] and stochastic[10] models given in Eqs. 9 and 10. The IMF with a causal influence identifies the key mechanism of the model data in stochastic (Fig. 2a) and deterministic (Fig. 2b) systems. These results indicate that the causal-decomposition method is suitable for separating causal interactions not only in the stochastic system, but also in the deterministic model where non-separability is generally assumed in the state space. Furthermore, we validated and compared the causal decomposition with existing causality methods in uncorrelated white noise with varying lengths, showing the consistency of causal decomposition in a short time series and under conditions where no causal interaction should be inferred (Fig. 3a). In addition, we assessed the effect of downsampling (Fig. 3b) and temporal shift (Fig. 3c) of a time series on causal decomposition and existing methods, showing that causal decomposition is less vulnerable to spurious causality due to sampling issues[3] and is independent of temporal shift, which is significantly confounded with the predictive causality method[19].

**Validation of causal-decomposition analysis.** We generated 10,000 pairs of uncorrelated white noise time-series observations with varying lengths ($L = 10$–$1000$) and calculated causality based

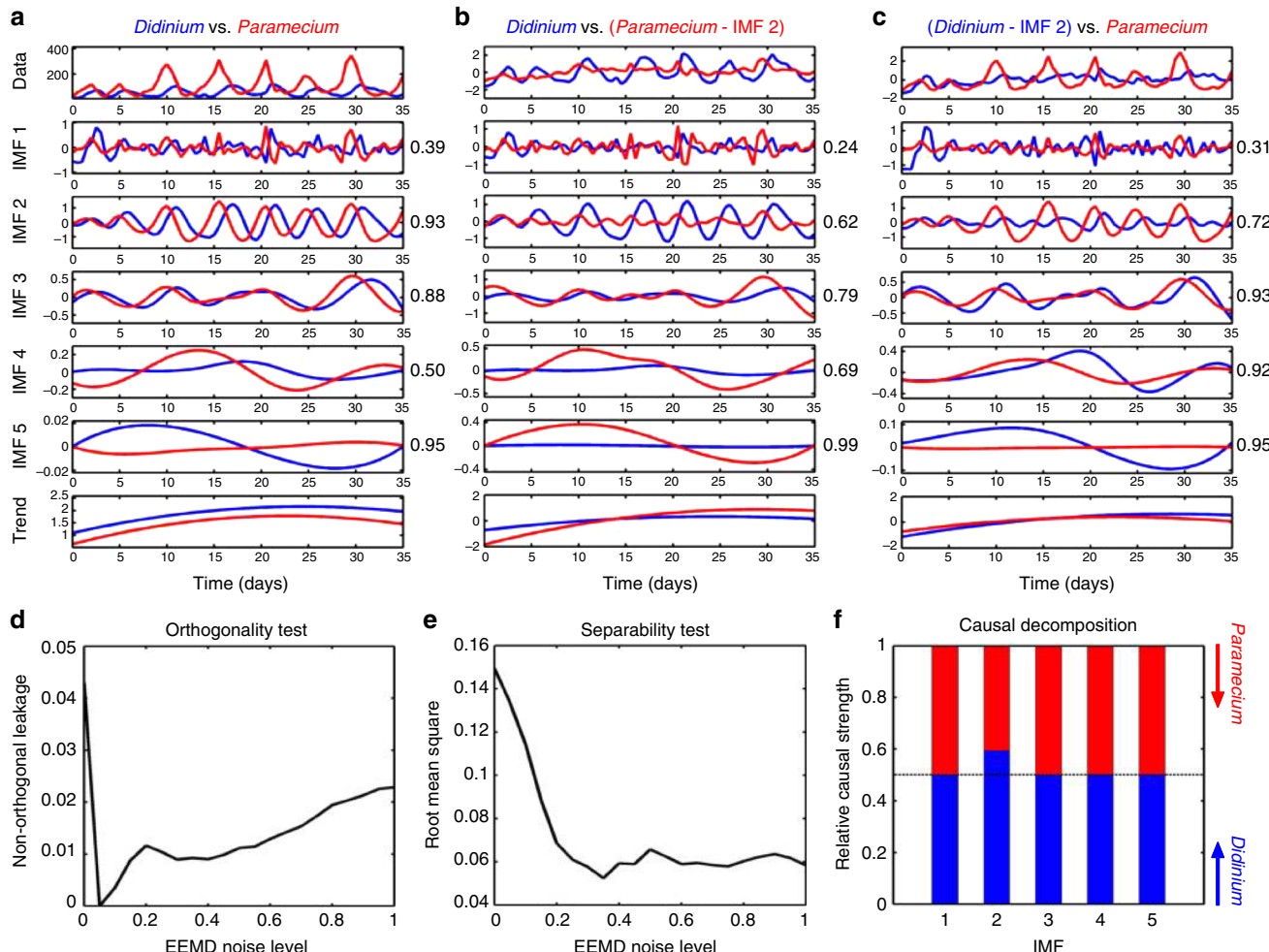

**Fig. 1** Causal-decomposition analysis. **a** Ensemble empirical mode decomposition (EEMD) analysis of *Didinium* (blue line) and *Paramecium* (red line) time series yields five Intrinsic Mode Functions (IMFs) (i.e., stationary components) and a residual trend (i.e., non-stationary trend). Each IMF operated at distinct time scales. Phase coherence values between comparable IMFs are shown at the right side of the panel. **b** Removal of an IMF (e.g., IMF 2) from *Paramecium* with redecomposition leads to a decreased phase coherence between the original *Didinium* IMFs and redecomposed *Paramecium* IMFs. **c** Repeating the same procedure in the *Didinium* time series resulted in a smaller decrease in phase coherence between the redecomposed *Didinium* IMFs and the original *Paramecium* IMFs. The causal strengths between *Didinium* and *Paramecium* can be estimated by the relative ratio of variance-weighted Euclidian distance of the phase coherence between (**b**) and **a** (for *Didinium*), and between **c** and **a** (for *Paramecium*). The ability of EEMD to separate time series depends on the orthogonality and separability of the IMFs with added noise, which can be evaluated by (**d**) nonorthogonal leakages and **e** the root-mean-square of correlations between pairwise IMFs. The strategy of choosing the added noise level in the EEMD is to maximise the separability (minimise the root-mean-square of pairwise correlation values among IMFs <0.05) while maintaining acceptable nonorthogonal leakages (<0.05). A noise level $r$ at 0.35 standard deviations of the time series was used in this case. **f** Generalisation of causal decomposition to each IMF uncovers a causal relationship from *Didinium* (blue bar) to *Paramecium* (red bar) in IMF 2 but not in the other IMFs, indicating a time scale-dependent causal interaction in the predator–prey system

on various methods (Fig. 3a). Causal decomposition exhibited a consistent pattern of causal strengths at 0.5 (the error bar denotes the standard error of causality assessment here and in the other panels), indicating that no spurious causality was detected, even in the case of the short noise time series. Causality in the CCM methods was indicated by the difference in correlations obtained from cross-mapping the embedded state space. In the case of uncorrelated white noise, the difference of correlation should be approximately zero, indicating no causality. However, the CCM method detects spurious causality with differences of up to 0.4 in the crossmap correlations in the short time series, and the difference between the correlations decreased as the signal length increased. A high percentage or intensity of spurious causality was also observed in Granger's causality and mutual information from the mixed embedding (MIME) method[20].

Next, we assessed the effect of down-sampling on the various causality methods (Fig. 3b). The stochastic and deterministic models shown in Fig. 2 are used (the corresponding colour for each variable is shown in the figure). The time series were down-sampled by a factor 1 to 10. For Factor 1, the time series were identical to the original signals. The down-sampling procedure destroyed the causal dynamics in both models and made causal inference difficult in predictive causality analysis[19]. Causal-decomposition analysis revealed a consistent pattern of the absence of causality when the causal dynamics were destroyed as the down-sampling factor was >2. However, spurious causality was detected with the predictive causality methods when the signals were down-sampled.

Finally, we evaluated the effect of temporal shift on the causality measures (Fig. 3c). Temporal shift (both lagged or

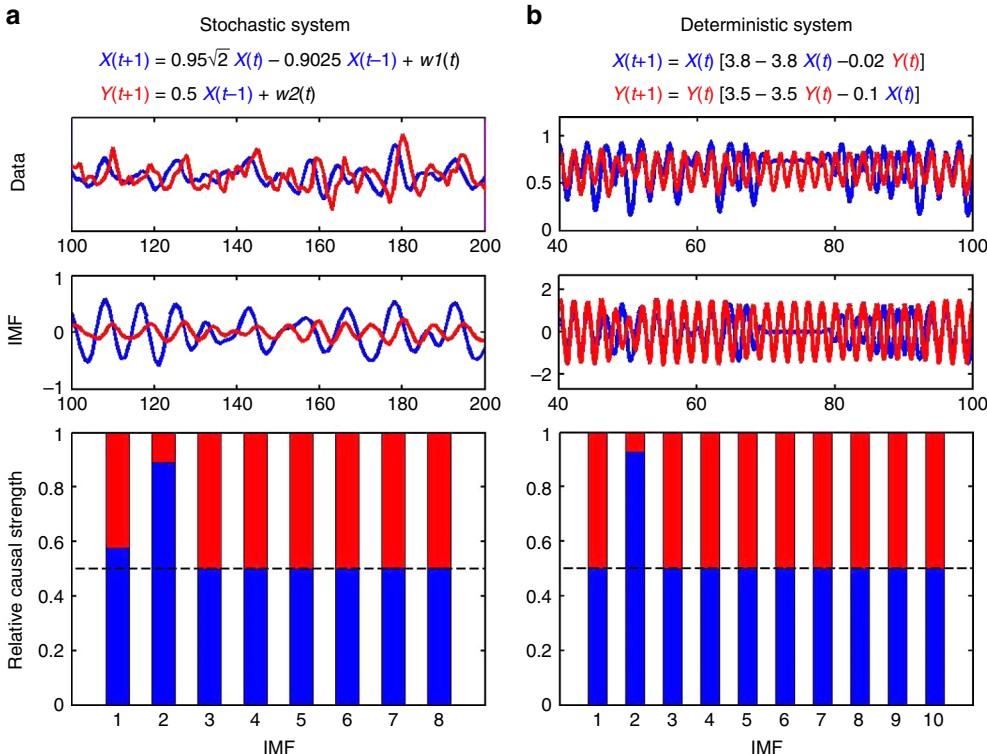

**Fig. 2** Stochastic and deterministic model evaluation. Application of causal decomposition to **a** stochastic system[10] and **b** deterministic system[5] (ensemble empirical mode decomposition; EEMD parameter $r = 0.15$ for both cases). A causal influence was identified in intrinsic mode function (IMF) 2 in both systems, capturing the main mode of signal dynamics in each system (e.g., a lag order of 2 between the IMFs in **a**, and chaotic behaviour of the logistic model in **b**). The causal decomposition is not only able to handle noisy data in the stochastic model, but it can also identify causal components in the deterministic model with the aid of EEMD in separating weakly coupled chaotic signals into identifiable IMFs. Data lengths: **a** 1000 data points; **b** 400 data points

advanced up to 20 data points) was applied to both the stochastic and deterministic time series. Causal decomposition exhibited a stable pattern of causal strength independent of a temporal shift up to 20 data points. CCM reduced its crossmap ability to detect causa interaction in the bi-directional deterministic system as temporal shift increased in either direction, and is unable to show differences in crossmap ability in the anterograde temporal shift in stochastic system. As anticipated, Granger's causality showed the opposite patterns of causal interaction in anterograde and retrograde temporal shift in both deterministic and stochastic system. MIME lost its predictability when the temporal shift is beyond 5 data points and was inconsistent in stochastic system.

**Quantifying predator and prey relationship.** Figure 4 shows the results of applying causal decomposition to ecosystem data from the Lotka Volterra predator–prey model[21,22] (Eq. 11; Fig. 4a), wolf and moose data from Isle Royale National Park[23] (Fig. 4b), and the Canada lynx and snowshoe hare time series reconstructed from historical fur records of Hudson's Bay Company[24] (Fig. 4c). The causal decomposition invariantly identifies the dominant causal role of the predator in the IMF, which is consistent with the classic Lotka Volterra predator–prey model. Previously, the causality of such autonomous differential equation models was understood only in mathematical terms because there is no prediction-based causal factor[25], yet our results indicated that the causal influence of this model can be established through the decomposition of instantaneous phase dependency.

**Comparison of causal assessment in ecosystem data.** Figure 5 shows the comparison of causality assessment in these predator and prey data using different methods. In general, results showed

that neither the Granger nor CCM methods consistently identify predator–prey interactions in these data, indicating that the predator–prey relationship does not exclusively fit either the stochastic or deterministic chaos paradigms. The CCM result showed a top–down causal interaction between lynx and hare, and *Didinium* and *Paramecium* interactions[17], which the latter was consistent with the data presented by Suigihara et al.[5] However, CCM method could not be used to detect causal interaction in the Lotka Volterra predator–prey model, and it exhibited a cross-over of correlations in the wolf and moose data. Granger's causality detected top–down causal interaction in the Lotka Volterra predator–prey model and wolf and moose data, but the bottom–up causal interaction was observed in *Didinium* and *Paramecium* data, which the latter was also observed in the supplementary data in Sugihara et al.[5] The inconsistency in causal strength was also observed in the results obtained with the MIME method.

## Discussion

An interdisciplinary problem of detecting causal interactions between oscillatory systems solely from their output time series has attracted considerable attention for a long time. The motivation of causal-decomposition analysis is that the inference of causality that is largely dependent on the temporal precedence principle is of concern. In other words, observing the past with a limited period is insufficient to infer causality because that history is already biased. Instead, we followed another fundamental criterion of causal assessment proposed by Galilei[1]—covariation of cause and effect: cause is that which put, the effect follows; and removed, the effect is removed. In this statement, however, the prediction of time series based on the past history is neither

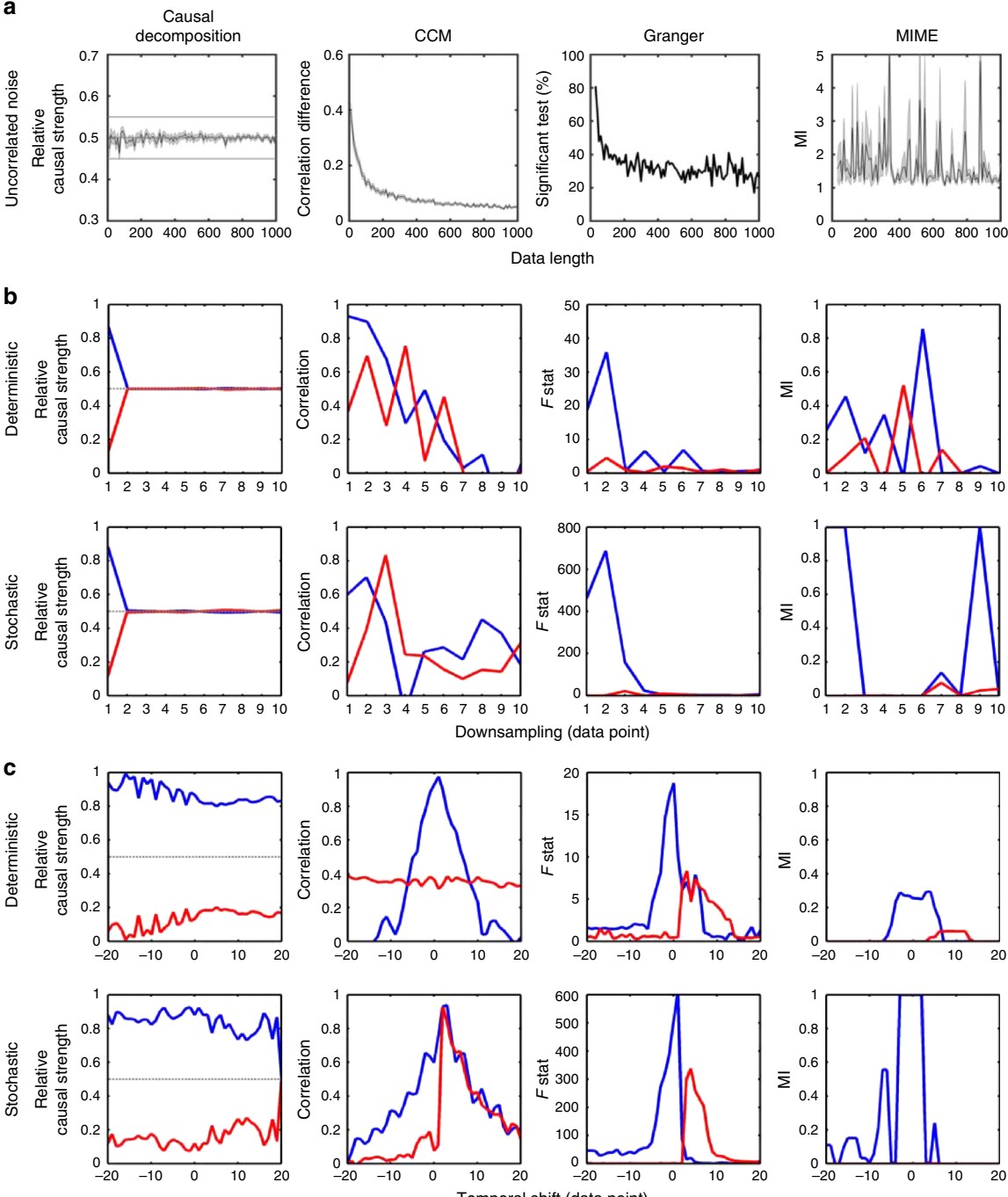

**Fig. 3** Validation of causal-decomposition method. **a** The finite length effect on causality assessment. We generated 10,000 pairs of uncorrelated white noise time-series observations with varying lengths ($L = 10$–1000) and calculated causality based on causal decomposition, convergent cross mapping (CCM), Granger causality, and mutual information from mixed embedding (MIME) method[20]. Causal decomposition exhibited a consistent pattern of causal strengths at 0.5 (the shaded error bar denotes the standard error of causality assessment here and in the other panels), indicating that no spurious causality was detected, whereas spurious causality was observed in the CCM, Granger's causality, and MIME method. **b** Effect of down-sampling on the various causality methods. The stochastic and deterministic models shown in Fig. 2 are used (the corresponding colour for each variable is shown in the figure). The time series were down-sampled by a factor 1 to 10. The down-sampling procedure destroyed the causal dynamics in both models and made causal inference difficult in predictive causality analysis[19]. Causal-decomposition analysis revealed a consistent pattern of the absence of causality when the causal dynamics were destroyed as the down-sampling factor was >2. **c** Effect of temporal shift on the various causality methods. Temporal shift (both lagged or advanced up to 20 data points) was applied to both the stochastic and deterministic time series. Causal decomposition exhibited a stable pattern of causal strength independent of a temporal shift up to 20 data points, while the predictive causality methods are sensitive to temporal shift

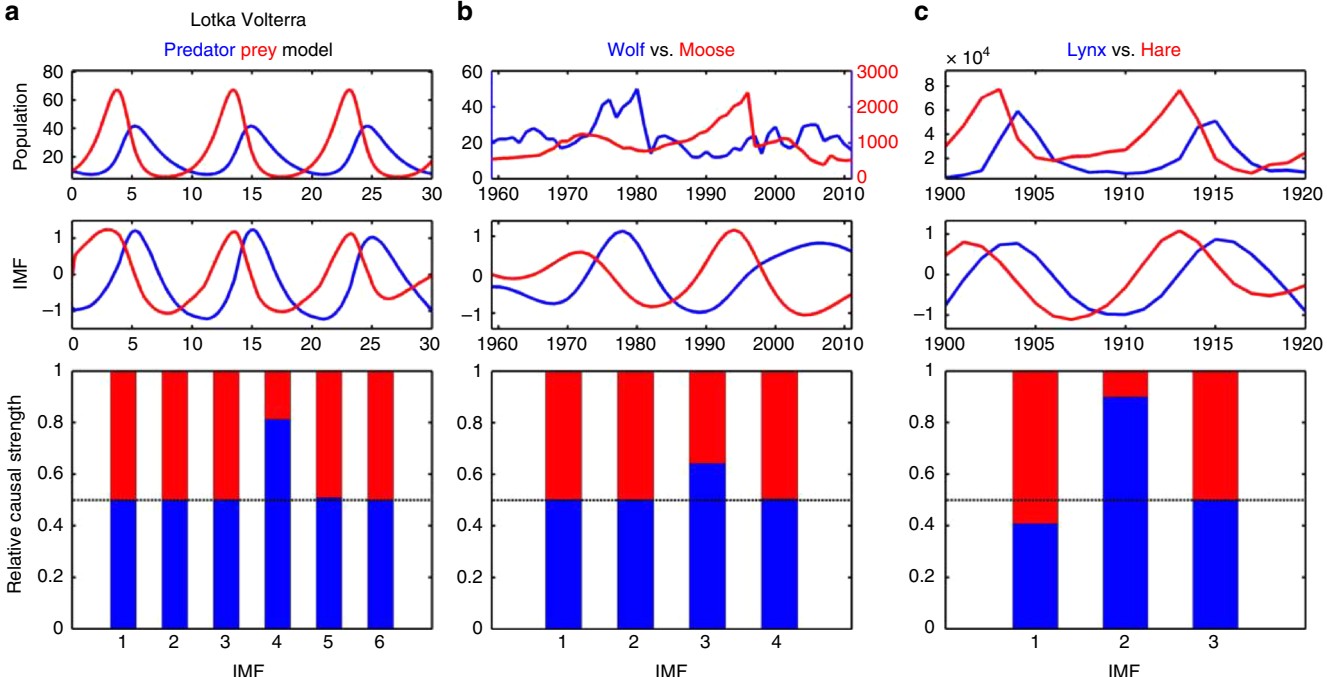

**Fig. 4** Causal decomposition of predator–prey data. **a** Lotka Volterra predator–prey model. **b** Wolf and moose time series from Isle Royale National Park in Michigan, USA[23]. **c** Canada lynx and snowshoe hare time series reconstructed from historical fur records of Hudson's Bay Company[24]. The intrinsic mode functions (IMFs) shown in the figure correspond to significant causal interactions identified in each observation (**a** IMF 4, **b** IMF 3, **c** IMF 2). Ensemble EMD parameter: **a** $r = 0.4$, **b** $r = 0.3$, **c** $r = 0.3$

required or implied. Therefore, the complex dynamical process between cause and effect should be delineated through the decomposition of intrinsic causal components inherited in causal interactions.

It is noteworthy that our approach is essentially different by combing EMD with existing causality methods, such as assessing Granger's causality between paired IMFs of economic time series[26], applying CCM to detect the nonlinear coupling of decomposed brain wave data[27], or measuring time dependency between IMFs decomposed from stock market data[28]. The decomposition of time series with EMD alone may improve the separability of intrinsic components embedded in the time series data, but does not avoid the constraints inherited from the existing prediction-based causality methods. Furthermore, our approach does not neglect the temporal precedence principle, but emphasises the instantaneous relationship of causal interaction, and is thus more amenable to detecting simultaneous or reciprocal causation, which is not fully accounted for by predictive methods.

Because our causal strengths measurement is relative, it detects differential causality rather than absolute causality. Differential causality adds to the philosophical concept of mutual causality that all causal effects are not equal, and it may fit the emerging research data better than linear and unidirectional causal theories do. In addition, causal decomposition using EMD fundamentally differs from the spectral extension of Granger's causality[29] in that the latter involves the prior knowledge of history (e.g., auto-regressive model order) and is susceptible to non-stationary artefacts. Furthermore, without resorting to frequency-domain decomposition, EMD bypasses the linear and stationary assumptions, and the limitation of uncertainty principle imposed on data characteristics as in Fourier analysis, and results in more precise phase and amplitude definition[30].

The operational definition of causal decomposition is in accordance with Granger's assumption on separability[3] but in a

more complete form. We note that such definition is distinct from non-separability assumed by CCM. Clearly, CCM is developed under the constraints of perfect deterministic system, in which the state of cause is encoded in effect that is not separable from effect itself. The state-space reconstruction approach such as CCM may be applicable to certain ecosystem data, such as predator and prey interactions, in which they represent non-separable components of the ecosystem[31], but is unlikely to generalise to all causal interactions being studied[32]. It is noteworthy that the effect of temporal shift on the CCM shown in Fig. 3c is relevant to the extended CCM to detect time-delayed causal interactions[33]. The extended CCM has been shown to capture bi-directional causal interactions in the deterministic system. However, in the real-world data, the time-delayed causal interaction has to be achieved by the arbitrary temporal shift of time series data, and the interpretation of such results is still of concern, as demonstrated in our Fig. 3c.

Several limitations should be considered in interpreting the causal strength presented in this paper. First, the causal decomposition represents a form of statistical causality and does not imply the true causality, which requires the inclusion of all variables to conclude the existence of causal relationship[3]. Second, the causal decomposition is limited to the pairwise measurement in the current form, but we do not exclude the possibility of the extension of the current method to multivariate systems (e.g., functional brain networks) with the employment of multivariate EMD[34,35] in the future. In that case, we have to define and work with the absolute causal strength matrix. Then the redecomposition would be from one to many. Although the causal principle remains the same, the computation would be time consuming.

The use of EMD overcomes the difficulty of signal decomposition in nonlinear and non-stationary data, and it is applicable to both stochastic and deterministic systems in that the intrinsic components in the latter remain separable in the time domain.

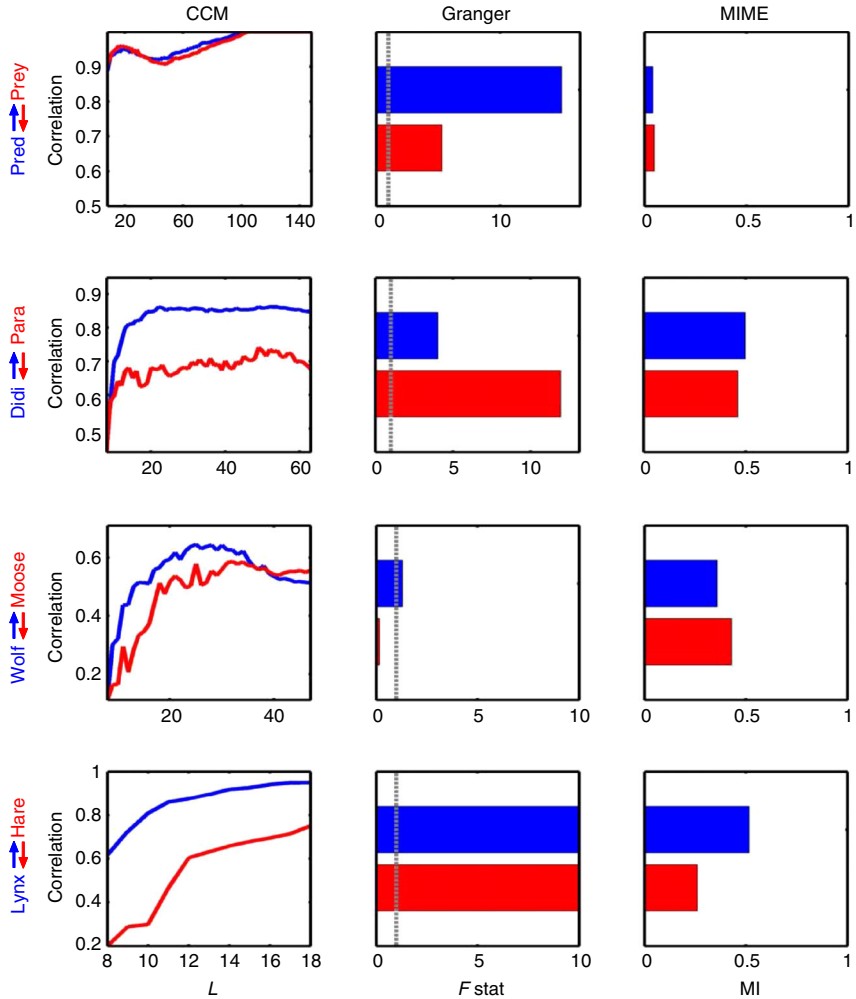

**Fig. 5** Comparison of causal assessment in ecosystem data with existing methods. The ecosystem data include Lotka Volterra predator–prey model[21,22] (first row), *Didinium* and *Paramecium* data[45] (second row), wolf and moose data from the United States Isle Royale National Park[23] (third row), and lynx and hare data from trading records obtained from Hudson's Bay Company[24] (fourth row). The results were derived from convergent cross mapping (CCM) (embedding dimension = 3), Granger's and mutual information from mixed embedding (MIME) methods. The colour of lines and bars indicate the causal strength of a given predator (blue) or prey (red). In CCM, the difference in correlation values between predator and prey indicates the direction of causal direction. In Granger causality, the *F*-test was used to assess the causal strength and the vertical dashed line denotes the significance threshold with *P* < 0.05. In MIME method, the relative causal strength was represented by difference in mutual information

Furthermore, the central element in causal-decomposition analysis is the decomposition and redecomposition procedure, and we do not exclude the use of other signal decomposition methods[36] to detect causality in a similar manner. Therefore, the development of causal decomposition is not to complement existing methods, but to explore the use of covariation principle of cause and effect for assessing causality. With the potential of the extension of ensemble EMD to multivariate EMD[34,35], we anticipate that this causal decomposition approach will assist with revealing causal interactions in complex networks not accounted for by current methods.

## Methods

**Causal relationship based on instantaneous phase dependency.** We define the cause–effect relationship between Time Series *A* and Time Series *B* according to the fundamental criterion of causal assessment proposed by Galilei[1]: cause is that which put, the effect follows; and removed, the effect is removed; thus, variable *A* causes variable *B* if the instantaneous phase dependency between *A* and *B* is diminished when the intrinsic component in *B* that is causally related to *A* is removed from *B* itself, but not vice versa.

$$\mathrm{Coh}(A, B') < \mathrm{Coh}(A, B) \sim \mathrm{Coh}(A', B) \qquad (1)$$

where Coh denotes the instantaneous phase dependency (i.e., coherence) between the intrinsic components of two time series, and the accent mark represents the time series where the intrinsic components relevant to cause effect dynamics were removed. The realisation of this definition requires two key treatments of the time series. First, the time series must be decomposed into intrinsic components to recover the cause–effect relationship at a specific time scale and instantaneous phase. Second, a phase coherence measurement is required to measure the instantaneous phase dependency between the intrinsic components decomposed from cause–effect time series.

**Empirical mode decomposition.** To achieve this, we decompose a time series into a finite number of IMFs by using the ensemble EMD[14–16] technique. Ensemble EMD is an adaptive decomposition method originated from EMD (i.e., the core of Hilbert–Huang Transform) for separating different modes of frequency and amplitude modulations in the time domain[14,15].

Briefly, EMD is implemented through a sifting process to decompose the original time-series data into a finite set of IMFs. The sifting process comprises the following steps: (1) connecting the local maxima or minima of a targeted signal to form the upper and lower envelopes by natural cubic spline lines; (2) extracting the first prototype IMF by estimating the difference between the targeted signal and the mean of the upper and lower envelopes; and (3) repeating these procedures to produce a set of IMFs that were represented by a certain frequency–amplitude modulation at a characteristic time scale. The decomposition process is completed when no more IMFs could be extracted, and the residual component is treated as

the overall trend of the raw data. Although IMFs are empirically determined, they remain orthogonal to one another, and may therefore contain independent physical meanings[15,37].

The IMF decomposed from EMD enables us to use Hilbert transform to derive physically meaningful instantaneous phase and frequency[14,29]. For each IMF, they represent narrow-band amplitude and frequency-modulated signal $S(t)$, and can be expressed as

$$S(t) = A(t)\cos\emptyset(t) \tag{2}$$

where instantaneous amplitude $A$ and phase $\emptyset$ can be calculated by applying the Hilbert transform, defined as $S_H = \frac{1}{\pi}\int \frac{S(t')}{t-t'}dt'$; $A(t) = \sqrt{S^2 + S_H^2}(t)$; and $\emptyset(t) = \arctan\left(\frac{S_H(t)}{S(t)}\right)$. The instantaneous frequency is then calculated as the derivative of the phase function $\omega(t) = d\emptyset(t)/dt$.

Thus, the original signal $X$ can be expressed as the summation of all IMFs and residual $r$,

$$X(t) = \sum_{j=1}^{k} A_j(t) \exp\left(i \int \omega_j(t)dt\right) + r \tag{3}$$

where $k$ is the total number of IMFs, $A_j(t)$ is the instantaneous amplitude of each IMF; and $\omega_j(t)$ is the instantaneous frequency of each IMF. Previous literature have shown that IMFs derived with EMD can be used to delineate time dependency[38] or phase dependency[37,39–42] in nonlinear and non-stationary data.

The ensemble EMD[15,16,43] is a noise-assisted data analysis method to further improve the separability of IMFs during the decomposition and defines the true IMF components $S_j(t)$ as the mean of an ensemble of trials, each consisting of the signal plus white noise of a finite amplitude.

$$S_j(t) = \lim_{N\to\infty} \sum_{k=1}^{N} \left\{ S_j(t) + r \times w_k(t) \right\} \tag{4}$$

where $w_k(t)$ is the added white noise, and $k$ is the $k$th trial of the $j$th IMF in the noise-added signal. The magnitude of the added noise $r$ is critical to determining the separability of the IMFs (i.e., $r$ is a fraction of a standard deviation of the original signal). The number of trials in the ensemble $N$ must be large so that the added noise in each trial is cancelled out in the ensemble mean of large trials ($N = 1000$ in this study). The purpose of the added noise in the ensemble EMD is to provide a uniform reference frame in the time–frequency space by projecting the decomposed IMFs onto comparable scales that are independent of the nature of the original signals. With the ensemble EMD method, the intrinsic oscillations of various time scales can be separated from nonlinear and non-stationary data with no priori criterion on the time–frequency characteristics of the signal. Hence, the use of ensemble EMD could complement the constraints of separability in Granger's paradigm[44] and potentially capture simultaneous causal relationships not accounted for by predictive causality methods.

**Orthogonality and separability of IMFs**. Because $r$ is the only parameter involved in the causal-decomposition analysis, the strategy of selecting $r$ is to maximise the separability while maintaining the orthogonality of the IMFs, thereby avoiding spurious causal detection resulting from poor separation of a given signal. We calculated the nonorthogonal leakage[14] and root-mean-square (RMS) of the pairwise correlations of the IMFs for each $r$ with an increment of 0.05 in the uniform space between 0.05 and 1. A general guideline for selecting $r$ in this study is to minimise the RMS of the pairwise correlations of the IMFs (ideally under 0.05) while maintaining the nonorthogonal leakage also under 0.05.

**Phase coherence**. Next, the Hilbert transform is applied to calculate the instantaneous phase of each IMF and to determine the phase coherence between the corresponding IMFs of two time series[18]. For each corresponding pair of IMFs from the two time series, denoted as $S_{1j}(t)$ and $S_{2j}(t)$, and can be expressed as

$$S_{1j}(t) = A_{1j}(t)\cos\emptyset_{1j}(t) \text{ and } S_{2j}(t) = A_{2j}(t)\cos\emptyset_{2j}(t), \tag{5}$$

where $A_{1j}$, $\emptyset_{1j}$ can be calculated by applying the Hilbert transform, defined as $S_{1jH} = \frac{1}{\pi}\int \frac{S_{1j}(t')}{t-t'}dt'$, and $A_{1j}(t) = \sqrt{S_{1j}^2 + S_{1jH}^2}(t)$, and $\emptyset_{1j}(t) = \arctan\left(\frac{S_{1jH}(t)}{S_{1j}(t)}\right)$; and similarly applied for $S_{2jH}$, $A_{2j}$, and $\emptyset_{2j}$. The instantaneous phase difference is simply expressed as $\Delta\emptyset_{12j}(t) = \emptyset_{2j}(t)\emptyset_{1j}(t)$. If two signals are highly coherent, then the phase difference is constant; otherwise, it fluctuates considerably with time. Therefore, the instantaneous phase coherence Coh measurement can be defined as

$$\text{Coh}\left(S_{1j}, S_{2j}\right) = \frac{1}{T}\left| \int_0^T e^{i\Delta\emptyset_{12j}(t)}dt \right| \tag{6}$$

Note that the integrand (i.e., $e^{i\Delta\emptyset_{12j}(t)}$) is a vector of unit length on the complex plane, pointing toward the direction which forms an angle of $\Delta\emptyset_{12j}(t)$ with the $+x$

axis. If the instantaneous phase difference varies little over the entire signal, then the phase coherence is close to 1. If the instantaneous phase difference changes markedly over the time, then the coherence is close to 0, resulting from adding a set of vectors pointing in all possible directions. This phase coherence definition allows the instantaneous phase dependency to be calculated without being subjected to the effect of time lag between cause and effect (i.e., the time precedence principle), thus avoiding the constraints of time lag in predictive causality methods[10].

**Causal decomposition between two time series**. With the decomposition of the signals by ensemble EMD and measurement of the instantaneous phase coherence between the IMFs, the most critical step in the causal-decomposition analysis is again based on Galilei's principle: the removal of an IMF followed by redecomposition of the time series (i.e., the decomposition and redecomposition procedure). If the phase dynamic of an IMF in a target time series is influenced by the source time series, removing this IMF in the target time series (i.e., subtract an IMF from the original target time series) with redecomposition into a new set of IMFs results in the redistribution of phase dynamics into the emptied space of the corresponding IMF. Furthermore, because the causal-related IMF is removed, redistribution of the phase dynamics into the corresponding IMF would be exclusively from the intrinsic dynamics of the target time series, which is irrelevant to the dynamics of the source time series, thus reducing the instantaneous phase coherence between the paired IMFs of the source time series and redecomposed target time series. By contrast, this phenomenon does not occur when a corresponding IMF is removed from the source time series because the dynamics of that IMF are intrinsic to the source time series and removal of that IMF with redecomposition would still preserve the original phase dynamics from the other IMFs. Therefore, this decomposition and redecomposition procedure enables quantifying the differential causality between the corresponding IMFs of two time series.

Because each IMF represents a dynamic process operating at a distinct time scale, we treat the phase coherence between the paired IMFs as the coordinates in a multidimensional space, and quantify the variance-weighted Euclidean distance between the phase coherence of the paired IMFs decomposed from the original signals as well as the paired original and redecomposed IMFs, which are expressed as follows:

$$D\left(S_{1j} \to S_{2j}\right) = \left\{ \sum_{j=1}^{m} W_j \left[ \text{Coh}\left(S_{1j}, S_{2j}\right) - \text{Coh}\left(S_{1j}, S_{2j}'\right) \right]^2 \right\}^{\frac{1}{2}}$$

$$D\left(S_{2j} \to S_{1j}\right) = \left\{ \sum_{j=1}^{m} W_j \left[ \text{Coh}\left(S_{1j}, S_{2j}\right) - \text{Coh}\left(S_{1j}', S_{2j}\right) \right]^2 \right\}^{\frac{1}{2}} \tag{7}$$

$$W_j = \left( \text{Var}_{1j} \times \text{Var}_{2j} \right) / \sum_{j=1}^{m} \left( Var_{1j} \times Var_{2j} \right)$$

The range of D represents the level of absolute causal strength and is between 0 and 1. The relative causal strength between IMF $S_{1j}$ and $S_{2j}$ can be quantified as the relative ratio of absolute cause strength $D\left(S_{1j} \to S_{2j}\right)$ and $D\left(S_{2j} \to S_{2j}\right)$, expressed as follows:

$$C\left(S_{1j} \to S_{2j}\right) = D\left(S_{1j} \to S_{2j}\right) \Big/ \left[ D\left(S_{1j} \to S_{2j}\right) + D\left(S_{2j} \to S_{1j}\right) \right]$$

$$C\left(S_{2j} \to S_{1j}\right) = D\left(S_{2j} \to S_{1j}\right) \Big/ \left[ D\left(S_{1j} \to S_{2j}\right) + D\left(S_{2j} \to S_{1j}\right) \right]. \tag{8}$$

This decomposition and redecomposition procedure is repeated for each paired IMF to obtain the relative causal strengths at each time scale, where a ratio of 0.5 indicates either that there is no causal relationship or equal causal strength in the case of reciprocal causation, and a ratio toward 1 or 0 indicates a strong differential causal influence from one time series to another. To avoid a singularity when both $D\left(S_{1j} \to S_{2j}\right)$ and $D\left(S_{2j} \to S_{1j}\right)$ approach zero (i.e., no causal change in phase coherence with the redecomposition procedure), $D + 1$ is used to calculate the relative causal strength when both absolute causal strength $D$ values are <0.05.

In summary, causal decomposition comprises the following three key steps: (1) decomposition of a pair of time series $A$ and $B$ into two sets of IMFs (e.g., IMFs $A$ and IMFs $B$) and determining the instantaneous phase coherence between each paired IMFs; (2) removing an IMF in a given time series (e.g., time series $A$), performing the redecomposition procedure to generate a new set of IMFs (IMF $A'$) and recalculating the instantaneous phase coherence between the original IMFs (IMFs $B$) and redecomposed IMFs (IMFs $A'$); and (3) determining the absolute and relative causal strength by estimating the deviation of phase coherence from the phase coherence of the original time series (IMFs $A$ vs. IMFs $B$) to either of the redecomposed time series (e.g., IMFs $A'$ vs. IMF $B$).

**Validation of causal strength**. To validate the causal strength, a leave-one-sample-out cross-validation is performed for each causal-decomposition test. Briefly, we delete a time point for each leave-one-out test and obtain a distribution of causal strength for all runs where the total number of time points is <100, or a maximum of 100 random leave-one-out tests where the total number of time points was higher than 100. A median value of causal strength is observed.

**Deterministic and stochastic model data**. The deterministic model was used in accordance with Sugihara et al.[5] based on a coupled two-species nonlinear logistic difference system, expressed as follows (initial value $x(1) = 0.2$, and $y(1) = 0.4$):

$$x(t+1) = x(t)[3.8 - 3.8x(t) - 0.02y(t)]$$
$$y(t+1) = y(t)[3.5 - 3.5y(t) - 0.1x(t)] \tag{9}$$

For the stochastic model, we used part of the example shown in Ding et al.[10] for Granger causality, which is expressed as follows (using a random number as the initial value).

$$x(t+1) = 0.95\sqrt{2}x(t) - 0.9025x(t-1) + w_1(t)$$
$$y(t+1) = 0.5x(t-1) + w_2(t) \tag{10}$$

**Ecological data and validation**. We assessed the causality measures in both modelled and actual predator and prey systems. The Lotka Volterra predator–prey model[21,22] is expressed as follows:

$$dx/dt = \alpha x - \beta xy$$
$$dy/dt = \delta xy - \gamma y \tag{11}$$

where $x$ and $y$ denote the prey and the predator, respectively ($\alpha = 1$, $\beta = 0.05$, $\delta = 0.02$, $\gamma = 0.5$ were used in this study).

Experimental data on *Paramecium* and *Didinium* are available online[45], and these were obtained by scanning the graphics in Veilleux[17] and digitising the time series. Wolf and moose field data are available online at the United States Isle Royale National Park[23]. The lynx and hare data were reconstructed from fur trading records obtained from Hudson's Bay Company[24]. The benchmark time series[46] was reconstructed from various sources in two periods (the 1844–1904 data were reconstructed from fur records, whereas the 1905–1935 data were derived from questionnaires)[24]. We used the fur-record time series between the year 1900 and 1920 for illustrative purposes.

**Comparison with other causality methods**. We compared causal decomposition with CCM, Granger's causality, and MIME method[20]. The detail of the calculation of CCM[5], Granger causality[10], and MIME[20] has been documented in the literature. Of note, both the CCM and Granger causality involve the selection of lag order. In this paper, the lag order (i.e., embedding dimension) of 3 was chosen for the application of CCM method to the ecosystem data[5], and the lag order in the Granger causality was selected by the Bayesian information Criterion. The MIME is an entropy-based causality method which also employs the time precedence principle[47] and is equivalent to Granger's causality in certain conditions[48].

**Code availability**. The source code for the causal-decomposition analysis (including ensemble EMD (http://rcada.ncu.edu.tw/research1.htm)) is implemented in Matlab (Mathworks Inc., Natick, MA, USA), and the current version (causal-decomposition-analysis-v1.0) or any future versions of the codes will be available at GitHub.

**Data availability**. The *Didinium* and *Paramecium* data that support the findings of this study are available in http://robjhyndman.com/tsdldata/data/veilleux.dat. Wolf and moose field data are available online at the United States Isle Royale National Park. Lynx and hare data are available online at https://github.com/bblais/Systems-Modeling-Spring-2015-Notebooks/tree/master/data/Lynx%20and%20Hare%20Data.

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

## Acknowledgements

This work was supported by the Ministry of Science and Technology (MOST) of Taiwan (Grant MOST 101-2314-B-075-041-MY3; 104-2314-B-075-078-MY2). We thank Dr. Shih-Jen Tsai, Dr. Shuu-Jiun Wang, Dr. Susan Shur-Fen Gau, Dr. Ching-Po Lin, Dr. Chang-Wei Wu, and Dr. Zhaohua Wu for valuable discussions.

## Author contributions

A.C.Y. developed the causal-decomposition method, performed the computational analysis, and wrote the manuscript. C.K.P and N.E.H. gave critical comments and contributed to manuscript writing. All authors discussed the results and approved the manuscript.

## Additional information

**Competing interests:** The authors declare no competing interests.

