## [Peer Review File · Nature Communications]

Reviewers' comments:

Reviewer #1 (Remarks to the Author):

The authors propose a causality approach which is applicable to time series and goes beyond the predictive paradigm that is behind most causality procedures used in the literature. Their method is characterized by A) Relying on empirical mode decomposition (EMD), B) Using phase dynamics, C) Using a decomposition/recomposition procedure meant to untangle the effect of the source from the intrinsic phase dynamics of the target time series. The authors apply this method to deterministic and stochastic models and to "predator-prey" data.

There is a significant component of novelty in the proposed approach, and the range of application is huge. The authors make a convincing case and their results are compelling. Overall, I think that this study is worth of publication. The main weakness is a formal one: the paper needs some rewriting in order to give more emphasis to its novel aspects. The main claim to originality is the ingenious combination of ensemble EMD, phase coherence and the decomposition/recomposition approach. I don't know if this is compatible with this journal's guidelines, but I would suggest restructuring the Methods section, starting with a brief description of what is already known (EMD, phase coherence...) and then describing the novelty (quantification of causality by incorporation of phase dynamics into the decomposition/recomposition procedure) in more detail in an isolated section.

Additional comments/suggestions:

line 64: The authors are of course aware that sophisticated methods for determining model order in Granger causality exist. Reference to this as a drawback needs to be further qualified. (It's only a drawback if these models are imperfect or error-prone.)

line 69: Although the authors mention this later on, Granger Causality methods that are frequency-specific (e.g. PDC) are well known. This should be stated here.

line 94: The use of verb tense is not fully coherent throughout the manuscript. For instance, past tense use ("was") used here is confusing. This should be corrected.

lines 100-103: The link between the properties of EEMD and its suitability for capturing phase dependency is not clear. This is a critical point that should be properly explained.

lines 122-123: Redundant with respect to lines 111-112.

line 128: Which increment was used to test r within the span 0.05-1?

lines 180ff: The use of relative causal strength instead of an absolute value raises concerns about what happens in the null condition, i.e. no causality link. I would expect relative causal strength to assume arbitrary values between 0 and 1 due to noise only. It does not look to me as if the strategy described in lines 188-191, amounting to a mere shift, is helpful to avoid this problem. Later on (line 276ff), the authors use white noise time series to demonstrate that the values, indeed, do not stray away from 0.5 (at least optically speaking). The authors should discuss this further and they should explain if they conducted any quantitative test to claim that the pattern of causal strengths is "consistent". Incidentally, Figure 3a does not show any error bar.

line 231: It's very weird that the authors had to resort to scanning a 1976 paper for their data. The authors should consider using a different dataset or explain what makes this data so special that they chose to use them instead of more recent, digitally available datasets.

Figure legends should definitely be rewritten. I see no reason for the imbalance between the long explanations given for Fig 1 and the scarcity of information in Fig 5.

Finally, the authors may consider citing Sweeney-Reed CM, Nasuto SJ. A novel approach to the detection of synchronisation in EEG based on empirical mode decomposition. J Comput Neurosci 2007;23(1):79–111.

Reviewer #2 (Remarks to the Author):

Dear authors

thanks for submitting this paper. While there are some interesting ideas behind it, I think that the method is far from being a breakthrough, and suffers from possibly more and more profound limitations of Granger causality and CCM.

First of all it should be always stressed that we are talking about statistical causality, not true one, and that in this sense this is a prima facie measurement, confounded by other actors, measured or not. In this sense your method (and the CCM one) are fundamentally limited being pairwise.

You state as a lemma the fact that prediction based methods "may" overlook the simultaneous and reciprocal nature of interactions. This is a strong statement based on not much evidence. Several formulations of Granger causality and variance and probability based tools (Transfer Entropy etc) take into account simultaneous interactions. In particular, the state space formulation (<https://journals.aps.org/pre/abstract/10.1103/PhysRevE.91.040101>) is robust to bias and variance.

Empirical mode decomposition has been already applied to the detection of dynamical interactions

www.mdpi.com/2071-1050/9/12/2299/pdf
<https://arxiv.org/abs/1708.06586>
<https://www.ncbi.nlm.nih.gov/pubmed/28835251>
<https://www.ncbi.nlm.nih.gov/pubmed/26419400>
DOI 10.1515/cdbme-2016-0050

and several others. It's strange that you don't mention any of these, and even more so that you consider an EMD-based strategy so novel and different than Granger causal and the like.

The direct application of Hilbert transform has been proven possibly problematic, and a protophase-to-phase transformation could be in order, see for example

http://www.stat.physik.uni-potsdam.de/~mros/pdf/PhysRevE_optphase_long.pdf

<http://www.stat.physik.uni-potsdam.de/~mros/pdf/optphase.pdf>

Line 190: using $D+1$ when both D values are less than 0.05 seems like a strong and rather unjustified solution.

Using thousand of trials to quantify relations based on "instantaneous" phase seems like a contradiction, and greatly limits your applicability.

The applications that you propose are quite artificial, and the comparison with other methods seems a bit artificial and different to quantify, since the methods are so different, but more importantly they are all designed to measure the effect and not the underlying mechanism, so

even the "ground truth" oscillatory model does not seem appropriate to any comparison.

The first sentence of the paper is curious, jumping in time and in concepts, from the philosophic notion of causality to the Granger formulation (which was maybe unfortunately called causality, but as you say, it's a prediction tool). It's even a bit arrogant from your side to state that you're after the "true" causality here, when your model is as data driven as all the others.

The paper is in general poorly written, the sentences are too long and involved, and too often some sentences are left there as lemmas when they are only suppositions or interpretations.

Reviewer #3 (Remarks to the Author):

The authors study causality in both real ecological time series and mathematical models in which there is bidirectional causality, in predator-prey or competitive relationships. They use instantaneous phase coherence as a measure of a causal relationship between two variables. In order to determine phase coherence, they decompose the time series of the two variables by means of ensemble empirical mode decomposition (EEMD). This methodology decomposes the time series into orthogonal oscillatory components, or intrinsic mode functions (IMFs). A Hilbert transform is applied to calculate the instantaneous phase for each IMF. The coherence of the phase difference between the corresponding IMFs of the two time series can then be calculated. Then the influence of one time series, the source, on the other, the target, can be determined by removing an IMF from the target and redecimating the time series without that IMF. If removal of an IMF that is causally related to the target, followed by redecimation, results in phase coherence that is lower, that is an indication of causal influence of the source on the target. The authors' approach, causal decomposition, determines the relative causality between the two variables (whether causality is stronger in one direction than the other) rather than absolute values of causality, so 50-50 is the default when no causality is found.

Comments

The methodology of empirical model decomposition (EMD) is well established in the literature (Huang et al. (1998), as its enhancement of EEMD (Wu and Huang), in which white noise is added as part of a process to provide a uniform reference frame for the IMFs. What is novel here is the application of these methods to determining which mode plays the key role in causal interaction.

The application to the Didinium-Paramecium time series shows somewhat higher top-down causality of Didinium on its Paramecium prey (Figure 1). This is in agreement with Sugihara's (2012) finding, using a different approach that uses prediction of one time series by another in state space, convergent cross-mapping (CCM). However, it should be noted that Ye et al, (2015), using CCM with time delay, found the causality to be the same in both directions at the optimal time lag. I am not sure what the implication of that result is for the present study.

The application to a deterministic discrete-time competitive relationship (equations 7) show stronger effect of one variable, X, on the other, Y (Figure 2b), which is also in agreement with Sugihara (2012), in which stronger effect is shown by faster convergence of predicting for X than for Y. The authors' approach works equally well on a discrete-time predator-prey system to which stochasticity is added (Figure 2a).

The authors compare causal decomposition with three other methods of inferring causality; data length, downsampling (making the signal smaller), and temporal shift. The comparisons are very interesting. However, more explanation should be given concerning the meaning of some of the results of the three other methods. It should be noted that Ye et al. (2015, Scientific Reports) extends CCM to time-delayed causal interactions, which may be relevant to the temporal shift comparisons shown in Figure 3c.

Figure 4 shows relative causal strengths for the LV model and two well-known time series. The results agree with the general view that the top-down effect of the predator is dominant. The authors might want to apply their approach to the Rosenzweig-MacArthur predator-prey model, which is much more realistic than the LV.

Figure 5. Again the text and figure caption are very brief and don't explain the meaning of the Granger and MIME results well enough. More interpretation would be helpful here.

The causal decomposition method appears to work well at determining relative causal influences for pairs of variables, both in model systems and in time series data from real ecological systems. I believe this work is novel and is a methodology that will be of broad interest, as it has applicability in many areas of science. I would like to see a bit more discussion of the authors' note near the end of the manuscript that "this novel approach will assist with revealing causal interactions in complex network". They should outline how the approach, which is only demonstrated for pairs of variables here, might be extended to complex systems that might have a number of variables.

Minor:

Line 26. Change 'scale' to 'scales' and 'from the target' to 'from that of the target'

Line 56. Change 'the total' to 'the totality'

Line 71. Change 'blended with' to 'blending'

Line 172. Change 'operating at distinct time scales' to 'operating at a distinct time scale'

Line 317. Change 'Fig. 5 showed' to 'Fig. 5 shows'

Reviewer #1 (Remarks to the Author):

The authors propose a causality approach which is applicable to time series and goes beyond the predictive paradigm that is behind most causality procedures used in the literature. Their method is characterized by A) Relying on empirical mode decomposition (EMD), B) Using phase dynamics, C) Using a decomposition/recomposition procedure meant to untangle the effect of the source from the intrinsic phase dynamics of the target time series. The authors apply this method to deterministic and stochastic models and to "predator-prey" data.

1-1: There is a significant component of novelty in the proposed approach, and the range of application is huge. The authors make a convincing case and their results are compelling. Overall, I think that this study is worth of publication. The main weakness is a formal one: the paper needs some rewriting in order to give more emphasis to its novel aspects. The main claim to originality is the ingenious combination of ensemble EMD, phase coherence and the decomposition/recomposition approach. I don't know if this is compatible with this journal's guidelines, but I would suggest restructuring the Methods section, starting with a brief description of what is already known (EMD, phase coherence...) and then describing the novelty (quantification of causality by incorporation of phase dynamics into the decomposition/recomposition procedure) in more detail in an isolated section.

Response 1-1: We thank the Reviewer for encouraging and helpful comments. Indeed, the introduction to EMD is missing in our original submission. Following the suggestion of the reviewer, we have restructured the method section to give a concise introduction to EMD method and the rationale to use EMD as a mean to measure phase dependency between time series to make it easier for the readers. We now detailed our responses to each comment listed below.

Additional comments/suggestions:

1-2: line 64: The authors are of course aware that sophisticated methods for determining model order in Granger causality exist. Reference to this as a drawback needs to be further qualified. (It's only a drawback if these models are imperfect or error-prone.)

Response 1-2: The point is well taken. Indeed, selection of model order in Granger causality has been guided by certain methods, such as Akaike information criterion (AIC) or Bayesian information criterion (BIC). The model order selection can impact the results of Granger causality that too few lags would lead to a biased causality test because of the residual auto-correlation, whereas too many lags would result in potentially spurious causality.

In general, we did not mean to devalue the constraint of lag order in Granger's method. In fact, AIC and BIC are often helpful to guide the model order selection in Granger causality. However, we attempt to point out that the use of Granger causality is largely dependent on the lag order (e.g., Thornton D. L. and Batten. D. S 1984.), as also demonstrated in our Figure 3c that temporal shift could largely affect the results of Granger causality test.

We revised our manuscript in the following manner to address the Reviewer's concern.

1. The use of word "drawback" may be a bit strong and misleading, so we changed it to the neutral term "constraint" to indicate that lag order is the important parameter of the Granger's method.
2. In fact, the Granger causality test presented in this paper already adopted BIC as the method to select the lag order, we added this information in the newly added "Comparison with other causality methods" in the last section of the Methods.

1-3: line 69: Although the authors mention this later on, Granger Causality methods that are frequency-specific (e.g. PDC) are well known. This should be stated here.

Response 1-3: We revised the manuscript accordingly as follows: *"Time series commonly observed in nature, including those from physiologic system or spontaneous brain activity, contain oscillatory components within specific frequency bands^{10, 11}. Identification of frequency-specific causal interaction is essential to understand the underlying mechanism^{12,13}. However, the application of either linear Granger causality or the nonlinear CCM method alone is insufficient to accommodate the complex causal compositions typically observed in real-world data blending with stochastic and deterministic mechanisms."*

1-4: line 94: The use of verb tense is not fully coherent throughout the manuscript. For instance, past tense use ("was") used here is confusing. This should be corrected.

Response 1-4: Thanks for pointing out this grammatical error. We now corrected this error and check throughout the manuscript to increase the consistency of the verb tense.

1-5: lines 100-103: The link between the properties of EEMD and its suitability for capturing phase dependency is not clear. This is a critical point that should be properly explained.

Response 1-5: This is indeed a critical point. We revised the manuscript in two aspects to accommodate Reviewer's concern. First, it is necessary to describe the original EMD and Hilbert-Huang Transform in the first place. Therefore, we added a detailed and necessary information in the section of empirical mode decomposition to explain how IMFs were defined and derived from EMD and the application of Hilbert Transform to calculate instantaneous phase in each IMF.

Second, we managed to add citations (alone with the one suggested by the Reviewer in Point **1-11**) that EMD could be used to capture phase dependency. We also added the statement to stress the necessity for invoking the Ensemble EMD. These citations evaluated and demonstrated the suitability of EMD or EEMD to assess the phase relationship between physiologic signals in previous literature, including

1. Novak V*, Yang AC, Lepicovsky L, Goldberger AL, Lipsitz LA, Peng CK. Multimodal pressure-flow method to assess dynamics of cerebral autoregulation in stroke and hypertension. *BioMedical Engineering Online* 3:39 (2004).
2. Sweeney-Reed CM, Nasuto SJ. A novel approach to the detection of synchronisation in EEG based on empirical mode decomposition. *Journal of computational neuroscience* 23, 79-111 (2007).
3. Lo MT, Novak V, Peng CK, Liu Y, Hu K. Nonlinear phase interaction between nonstationary signals: a comparison study of methods based on Hilbert-Huang and Fourier transforms. *Physical review* 79, 061924 (2009).
4. Hu K, Lo MT, Peng CK, Liu Y, Novak V. A nonlinear dynamic approach reveals a long-term stroke effect on cerebral blood flow regulation at multiple time scales. *PLoS computational biology* 8, e1002601 (2012).
5. Cho D, Min B, Kim J, Lee B. EEG-Based Prediction of Epileptic Seizures Using Phase

Synchronization Elicited from Noise-Assisted Multivariate Empirical Mode Decomposition. IEEE transactions on neural systems and rehabilitation engineering : a publication of the IEEE Engineering in Medicine and Biology Society 25, 1309-1318 (2017).

In this revision we therefore cited these references as the support for applying Hilbert transform to IMFs to capture phase dependency between nonlinear and nonstationary signals.

1-6: lines 122-123: Redundant with respect to lines 111-112.

Response 1-6: We removed the redundant sentence in lines 122-123 accordingly. ~~“Additionally, the orthogonality and separability of IMFs are critically dependent on selecting the magnitude r of the added white noise in ensemble EMD.”~~

1-7 line 128: Which increment was used to test r within the span 0.05-1?

Response 1-7: We apologized for the missing information here. The increment was 0.05 and we had appended this information in the revised manuscript.

1-8: lines 180ff: The use of relative causal strength instead of an absolute value raises concerns about what happens in the null condition, i.e. no causality link. I would expect relative causal strength to assume arbitrary values between 0 and 1 due to noise only. It does not look to me as if the strategy described in lines 188-191, amounting to a mere shift, is helpful to avoid this problem. Later on (line 276ff), the authors use white noise time series to demonstrate that the values, indeed, do not stray away from 0.5 (at least optically speaking). The authors should discuss this further and they should explain if they conducted any quantitative test to claim that the pattern of causal strengths is "consistent". Incidentally, Figure 3a does not show any error bar.

Response 1-8: This is a good point. In fact, the absolute value does indicate the case of no causality link. Here we showed the same tests used in Figure 3a to generate 10 000 pairs of uncorrelated white noise time-series observations with varying lengths ($L = 10-1000$). Instead of reporting relative causal strength in the Figure 3a, here we showed the change of absolute causal values (i.e., $D(S1 \rightarrow S2)$ and $D(S2 \rightarrow S1)$) in the top of the next page.

Both figures suggest that the standard error of absolute causal strength between uncorrelated noise time series is very low and does not distribute randomly between 0 and 1. Furthermore, the absolute causal strength in either direction shows a similar trend, thereby results in a stable relative causal strength around 0.5 shown in Figure 3a across different data length. This observation suggests that the use of relative causal strength does have the advantage over the absolute causal strength in evaluating causality.

On the other hand, it is still possible to assess absence of causal link using absolute causal strength. Arbitrarily, if we set the threshold of absolute causal strength at the level of 0.05 to reject null condition (grey dashed line), it would require the data length to be at least 500 data points to obtain a stable measure of low absolute causal strength, instead of

In this revision, we made following revisions to address Reviewer's concern.

1. Figure 3a does have the error bar, the range of error was too small to clearly be seen on the scale of relative causal strength at 0 and 1. Also the error bars overlapped with each other and make the interpretation difficult. Therefore, we revised this figure to a) change the scale of Y-axis to be between 0.3 and 0.7 in the results of causal decomposition in Figure 3a; b) Additionally, we also changed the Y-axis to be between 0 and 0.6 in CCM results in Figure 3a. So it's easier to see the error bar in both figures; c) we use shaded error line plot to make the figure more clear; 4) we added a grey horizontal line at 0.45, 0.5, and 0.55 to indicate an arbitrary range of relative causal

strength.

2. We do note that in Figure 3, the label for causal decomposition method was “Causal Strength” and we changed the label in this revision to “Relative Causal Strength” so avoid potential confusion with absolute causal strength, D.

3. In terms of consistency, we also set an arbitrary threshold of relative causal strength to be 0.05 around 0.5 level (i.e., 0.45-0.55 to be considered as accepting null condition). We found that relatively causal strength between uncorrelated noise does show consistency across different data lengths.

1-9: line 231: It's very weird that the authors had to resort to scanning a 1976 paper for their data. The authors should consider using a different dataset or explain what makes this data so special that they chose to use them instead of more recent, digitally available datasets.

Response 1-9: It may seem odd to use data obtained in 1976. The main reason for using this set of data is that the same data was used to demonstrate the applicability of CCM methods by Dr. Sugihara et al. (Science 2012). We thought that the comparison of methods using the same data would be appropriate and necessary to make our point more clear.

1-10: Figure legends should definitely be rewritten. I see no reason for the imbalance between the long explanations given for Fig 1 and the scarcity of information in Fig 5.

Response 1-10: We examined all figure captions and appended necessary information for each figure in this revision.

1-11: Finally, the authors may consider citing Sweeney-Reed CM, Nasuto SJ. A novel approach to the detection of synchronization in EEG based on empirical mode decomposition. J Comput Neurosci 2007;23(1):79–111.

Response 1-11: We apologized for not citing this paper in the origin submission and added this as a reference to support the use of EMD to quantify phase dependency between physiologic signals (see also response to Point 1-5).

Reviewer #2 (Remarks to the Author):

Dear authors

thanks for submitting this paper. While there are some interesting ideas behind it, I think that the method is far from being a breakthrough, and suffers from possibly more and more profound limitations of Granger causality and CCM.

2-1: First of all it should be always stressed that we are talking about statistical causality, not true one, and that in this sense this is a prima facie measurement, confounded by other actors, measured or not. In this sense your method (and the CCM one) are fundamentally limited being pairwise.

Response 2-1: We acknowledged this as one of constraints of causality decomposition method and would like to add a limitation section to address these concerns. The newly added paragraph is as follows:

“Several limitations should be considered in interpreting the causal strength presented in this paper. First, the causal decomposition represents a form of statistical causality and does not imply the true causality, which requires the inclusion of all variables to conclude the existence of causal relationship (Granger 1969). Second, the causal decomposition is limited to the pairwise measurement in the current form, but we do not exclude the possibility of the extension of the current method to multivariate systems (e.g., functional brain networks) with the employment of multivariate EMD (Rehman N 2009, Zhang Y 2017) in the future.”

2-2: You state as a lemma the fact that prediction based methods "may" overlook the simultaneous and reciprocal nature of interactions. This is a strong statement based on not much evidence. Several formulations of Granger causality and variance and probability based tools (Transfer Entropy etc) take into account simultaneous interactions. In particular, the state space formulation (<https://journals.aps.org/pre/abstract/10.1103/PhysRevE.91.040101>) is robust to bias and variance.

Response 2-2: This is a good point but we think we do not miss the point here in the abstract. As pointed out in the Sugihara's 2012 CCM paper, the fundamental constraint

of Granger's method is separability, thus the predictive causality method like Granger's and its variants can underestimate the simultaneous and reciprocal nature of interaction when the coupling is weak or non-separable.

The incorporation of state space in Barnett and Seth's 2015 paper does improve the reliability of Granger causality in the case of filtered, downsampled, or noisy data (such as the case we showed in Figure 3b), but we didn't see the evidence presented in that paper could be generalized to the mutual causation system such as the predator and prey model.

We have shown in this paper the use of state space (such as CCM) or probability based causal tool (MIME) performed inconsistently in the case of predator and prey models (though state-space based CCM does perform better than Granger causality), and this can provide the conceptual evidence that the predictive methods could underestimate the reciprocal nature of interactions at least in predator and prey models.

We understand that the word "overlook" is strong and provocative. In response to Reviewer's comment, we have rephrased our sentence in the abstract by using the term "underestimate" instead of "overlook" to address the concern reasonably and precisely.

2-3: Empirical mode decomposition has been already applied to the detection of dynamical interactions

www.mdpi.com/2071-1050/9/12/2299/pdf

<https://arxiv.org/abs/1708.06586>

<https://www.ncbi.nlm.nih.gov/pubmed/28835251>

<https://www.ncbi.nlm.nih.gov/pubmed/26419400>

DOI 10.1515/cdbme-2016-0050

and several others. It's strange that you don't mention any of these, and even more so that you consider an EMD-based strategy so novel and different than Granger causal and the like.

Response 2-3: Thanks for the Reviewer to mention these references. Indeed, EMD has been applied to many problems including the detection of dynamical interactions, and we should cite some of these references. However, we would like to mention the

fundamental differences of our approaches with these studies.

1. Jiang L and Bai Ling, Revisiting the Granger Causality Relationship between Energy Consumption and Economic Growth in China: A Multi-Timescale Decomposition Approach. *Sustainability* 9, 2299 (2017).

This paper described the use of ensemble EMD to decompose the economic time series of energy consumption and economic growth into a set of IMFs and apply granger causality to assess the causality between paired IMFs. This approach simply used the Granger's method to assess causality in the intrinsic components of data but does not overcome the constraints of Granger's method. For example, we can apply Granger's causality to paired IMFs shown in Figure 1a of *Didinium* and *Paramecium* IMF data.

Table. Granger causality test to paired IMFs of *Didinium* and *Paramecium* data

IMF Order	Didinium -> Paramecium	Paramecium -> Didinium
1	0.27	12.23*
2	10.98*	20.33*
3	1.11	4.34*
4	4.43*	9.61*
5	15.24*	9.56*

Values are shown as F test value and asterisk indicates the significant Granger's test at the level of $P < 0.05$.

The table is consistent with what we have shown in Figure 5 that Granger's test cannot consistently infer causal interactions in the Predator and Prey model. Furthermore, this approach has a limitation that it assumes every IMF has physical meanings, which was not true in many cases. For example, we found only certain IMF has causal inferences but not in other IMFs in Figure 1, 2, and 4. Applying Granger causality to paired IMFs data will not prevent from observing spurious causality (as shown in the table above).

2. Nava, N., Di Matteo, T., Aste, T. Dynamic correlations at different time-scales with Empirical Mode Decomposition. *Physica A* 2018.

This paper applied EMD to assess the time dependency between IMFs decomposed from stock market data. The paper does not assess causality, though time dependency may

give some clues about led or lag relations but it's fundamentally different from what we proposed in our current work. We will cite this reference in the discussion.

3. Zhang Y. et al. Noise-assisted multivariate empirical mode decomposition for multichannel EMG signals. *Biomed Eng Online*. 2017; 23;16(1):107.

This paper compared the performance (e.g., spectral characteristics) of decomposition of EMG signal using noised-assisted multivariate EMD, multivariate EMD, or ensemble EMD. The paper does not involve the assessment of causality but we do think the multivariate approach could be implemented in our future study of causality in multivariate systems (such as functional brain networks), so we will cite this paper together with another paper of multivariate EMD (see also response **2-1**).

4. Schiecke K et al. Assignment of Empirical Mode Decomposition Components and Its Application to Biomedical Signals. *Methods Inf Med*. 2015;54(5):461-73. / Piper D et al. Comparative study of methods for solving the correspondence problem in EMD applications. *Current Directions in Biomedical Engineering*.

These two papers presented an algorithm to match corresponding IMFs in the multivariate data. This study does not involve the causality assessment and we do not think the results will be helpful to improve our work. Moreover, the correspondence problem has been addressed in the paper of ensemble EMD (Wu Z 2005). Therefore, we will not cite these two papers in our revised manuscript.

5. Additionally, during the literature search, we do find another paper combining CCM method with EMD algorithm (Schiecke K, et al. Advanced nonlinear approach to quantify directed interactions within EEG activity of children with temporal lobe epilepsy in their time course. *EPJ Nonlinear Biomed Phys* 2017; 5, 3).

Similar to the combined EMD/causality approach in the first paper (Jiang 2017) mentioned by the Reviewer, this paper applied CCM to study the nonlinear coupling between decomposed IMFs.

We think this approach will not avoid the constraints imposed in the CCM method. Using *Didinium* and *Paramecium* data as an example, we found that the largest crossmap causal difference (indicated by the difference in correlation value of blue and red lines)

was found in IMF3 (which is not the key mode of predator and prey relationship). CCM does detect some causal interactions in IMF 2 and 5 but the difference is small. In addition, CCM results do not make sense in other paired IMFs of *Didinium* and *Paramecium* data (e.g., IMF 1 and IMF 4; see figure below).

Fig. CCM results of paired IMFs of causal interactions from *Didinium* to *Paramecium* (blue) and *Paramecium* to *Didinium* (red) data.

In summary, we think that it's important to address the fundamental difference of our approach with other studies that using EMD in the study of dynamical features of time series data. Therefore, we added a brief discussion and cited some of papers suggested by the Reviewer in the revised manuscript to address Reviewer's concern as follows:

"It is noteworthy that our approach is essentially different with combining EMD with existing causality methods, such as assessing Granger's causality between paired IMFs of economic time series (Jiang 2009), applying CCM to detect nonlinear coupling of decomposed brain wave data (Schiecke 2017), or measuring time dependency between IMFs decomposed from stock market data (Nava 2018). The decomposition of time series with EMD alone may improve the separability of intrinsic components embedded in the time series data, but do not avoid the constraints inherited in the existing prediction-based causality methods."

2-4: The direct application of Hilbert transform has been proven possibly problematic, and a protophase-to-phase transformation could be in order, see for example http://www.stat.physik.uni-potsdam.de/~mros/pdf/PhysRevE_optphase_long.pdf
<http://www.stat.physik.uni-potsdam.de/~mros/pdf/optphase.pdf>

Response 2-4: The point is very well taken. The application of Hilbert Transform indeed requires certain constraints to work properly in the non-stationary time series. As pointed out in Kraleman's work, phase estimation from data via the Hilbert transform actually provide protophases which heavily depend on the scalar observables available and on the analysis technique.

However, it is noteworthy that the purpose of protophase-to-phase transformation is to reconstruct phase dynamics equations for coupled oscillator, which is not the goal or process involved in our study.

In causal decomposition method, the estimation of instantaneous phase is to calculate "phase locking" of IMFs. As pointed out in Kraleman's paper *"If the goal of the analysis is the quantification of the frequency locking, then the difference between phases and protophases is not relevant, since they provide the same average frequencies, However, if the goal of the analysis is to get more insight into the interaction and to reconstruct the equations of the phase dynamics, this difference becomes decisive."* Therefore, we think the direct application of Hilbert transform to IMFs to derive phase dependency is appropriate in our case.

Furthermore, we would like to emphasize that the purpose of EMD is to solve the limitation of Hilbert transform, as demonstrated in the Huang 1998 paper. For example, in the page 913 of Huang 1998 paper, Huang has explained that *"the bandwidth limitation on the Hilbert transform to give a meaningful instantaneous frequency has never been firmly established."* Therefore, he proposed to use intrinsic mode function (IMF) that satisfies two conditions *"(1) in the whole data set, the number of extrema and the number of zero crossings must either equal or differ at most by one; and (2) at any point, the mean value of the envelope defined by the local maxima and the envelope defined by the local minima is zero."* Huang has further pointed out in the next section that *"The first condition is obvious; it is similar to the traditional narrow band requirements for a stationary Gaussian process. The second condition is a new idea; it modifies the classical global requirement to a local one; it is necessary so that the*

instantaneous frequency will not have the unwanted fluctuations induced by asymmetric wave forms."

Because most of the raw data are not IMFs. At any given time, the data may involve more than one oscillatory mode; that is why the Hilbert transform cannot uncover phase dynamics without proper decomposition, and EMD was developed to resolve this issue to derive a mathematically meaningful instantaneous phase/frequency.

In response to the Reviewer's concern, we added a brief description in the newly added EMD section (see also response 1-5) to address that *"The IMF decomposed from EMD overcomes the limitation as imposed by Hilbert transform to derive physically meaningful instantaneous phase and frequency (Huang et al., 1998; Huang et al., 2009)."*

2-5: Line 190: using $D+1$ when both D values are less than 0.05 seems like a strong and rather unjustified solution.

Response 2-5: The reason we choose $D+1$ in low absolute causal strength less than 0.05 was based on the observation of causal strength in Figure 3a. As also shown in the response to Reviewer 1's comment (response 1-8), the absolute causal strength between uncorrelated noise time series remains symmetric across different data lengths but it can reduce to a low level after 500 data points, indicating that causal decomposition can reliably detect no causal interaction in terms of absolute causal strength. Therefore, to prevent the singularity in the calculation of relative causal strength that has no causal interactions (i.e., low absolute causal strength), we set the $D+1$ when absolute causal strength below the level of 0.05.

We revised the method section as follows: "The range of D represents the level of absolute causal strength and is between 0 and 1. To avoid a singularity when both $D(S_{1j} \rightarrow S_{2j})$ and $D(S_{2j} \rightarrow S_{1j})$ approach zero (i.e., no causal change in phase coherence with the redecomposition procedure), $D + 1$ was used to calculate the relative causal strength when both absolute causal strength D values are less than 0.05."

2-6: Using thousand of trials to quantify relations based on "instantaneous" phase seems like a contradiction, and greatly limits your applicability.

Response 2-6: If we understand the comment correctly, we would like to clarify that

multiple trials of added noise in the ensemble EMD is to improve the quality and separability of final IMFs, by averaging all trials of decomposition to cancel out the effect of noise. Each trail still preserves the temporal characteristic of the data; therefore, the ensemble of thousands of trails would not mask any relationship based on 'instantaneous' phase. In our study, the phase relationship is not derived from each trial of decomposition but directly from final IMFs in the ensemble EMD.

Furthermore, using ensemble EMD (or so-called noise-assisted EMD) is necessary in this method because the separability is the key to uncover causal inference. Original EMD method (Huang 1998) or other decomposition methods may suffer from mode-mixing problems which hampers the separability of intrinsic components during the decomposition.

2-7: The applications that you propose are quite artificial, and the comparison with other methods seems a bit artificial and different to quantify, since the methods are so different, but more importantly they are all designed to measure the effect and not the underlying mechanism, so even the "ground truth" oscillatory model does not seem appropriate to any comparison.

Response 2-7: We understand that the Reviewer felt that our approach pursuit for "true" causality, which we believe we never made such a claim in our original submission. We would like to recap the modern definition of causality here, which includes 1) time precedence principle (i.e., cause precedes effect), 2) covariation of the cause and the effect (i.e., if cause then effect, if no cause then no effect), and 3) no plausible alternative explanations.

The causal decomposition is indeed different with others because it involves primarily the second principle (i.e., covariation of the cause and effect), while the majority of existing methods are based on the first principle (time precedence). This is why our method can reliably catch the causal inference in the case of time shifts as demonstrated in Figure 3. And based on our results, we think the second principle (covariation of cause and effect) is a new addition to causal analysis in the time series data.

We do concur that all current methods, including the one we proposed, have the same constraint of being statistical causality but not a true one unless all plausible variables have been tested, and we think that we have emphasized this in our revised manuscript.

We think the application and the comparison are not artificial, for all the examples are used in classical studies of causality. Therefore, our study does represent the state-of-art on the causality studies. Our results show the major difference between various causality methods and their constraints quantitatively, which are important to help understand the pros and cons and differences between methods.

2-8: The first sentence of the paper is curious, jumping in time and in concepts, from the philosophic notion of causality to the Granger formulation (which was maybe unfortunately called causality, but as you say, it's a prediction tool). It's even a bit arrogant from your side to state that you're after the "true" causality here, when your model is as data driven as all the others.

Response 2-8: In accordance with our response 2-7, we never claim or imply that our results represent a true causality. We simply followed the covariance principle to assess causal inference in the time series.

We understand that there has been a serious debate after Granger presented and called his method as "causality". We would like to point out that Granger and others' approach, including ours, do follow a certain principle of causality. Such is the general consensus of the community. Here, our goal is to present a different approach to the study of causality in the time series as accepted in the research community. We did examine the paper and modified some words usage to avoid any misleading terms.

2-9: The paper is in general poorly written, the sentences are too long and involved, and too often some sentences are left there as lemmas when they are only suppositions or interpretations.

Response 2-9: We thank the Reviewer for critical comments on our writing style. These comments did instigate us to rethink and revise our manuscript to the best of our ability. We managed to revise the manuscript to improve the clarity of the method and explained the major differences between the existing methods and the one we proposed here. We hope that the revised manuscript can be of help to address concerns raised by the Reviewer and make our main points across to the readers.

Reviewer #3 (Remarks to the Author):

3-1: The authors study causality in both real ecological time series and mathematical models in which there is bidirectional causality, in predator-prey or competitive relationships. They use instantaneous phase coherence as a measure of a causal relationship between two variables. In order to determine phase coherence, they decompose the time series of the two variables by means of ensemble empirical mode decomposition (EEMD). This methodology decomposes the time series into orthogonal oscillatory components, or intrinsic mode functions (IMFs). A Hilbert transform is applied to calculate the instantaneous phase for each IMF. The coherence of the phase difference between the corresponding IMFs of the two time series can then be calculated. Then the influence of one time series, the source, on the other, the target, can be determined by removing an IMF from the target and redecomposing the time series without that IMF. If removal of an IMF that is causally related to the target, followed by redecomposition, results in phase coherence that is lower, that is an indication of causal influence of the source on the target. The authors' approach, causal decomposition, determines the relative causality between the two variables (whether causality is stronger in one direction than the other) rather than absolute values of causality, so 50-50 is the default when no causality is found.

Response 3-1: The summary of our method is perfect. Thank you!

Comments

3-2: The methodology of empirical model decomposition (EMD) is well established in the literature (Huang et al. (1998), as its enhancement of EEMD (Wu and Huang), in which white noise is added as part of a process to provide a uniform reference frame for the IMFs. What is novel here is the application of these methods to determining which mode plays the key role in causal interaction.

Response 3-2: Indeed, the ensemble EMD is the key to uncover the key mode of causal interaction by separating intrinsic components of data above and beyond the other modes

3-3: The application to the Didinium-Paramecium time series shows somewhat higher

top-down causality of *Didinium* on its *Paramecium* prey (Figure 1). This is in agreement with Sugihara's (2012) finding, using a different approach that uses prediction of one time series by another in state space, convergent cross-mapping (CCM). However, it should be noted that Ye et al, (2015), using CCM with time delay, found the causality to be the same in both directions at the optimal time lag. I am not sure what the implication of that result is for the present study.

Response 3-3: This is a very important suggestion and we apologize not citing this paper about extended CCM in our original submission. We studied Ye et al paper in depth. First, we attempted to replicate the findings shown in the paper. For example, we are able to replicate the finding of *Didinium* and *Paramecium* data as shown below:

To simplify the presentation, the blue line here indicates top-down causal interaction from *Didinium* to *Paramecium* (i.e., *Paramecium* crossmap *Didinium*), and red line indicates bottom-up interaction from *Paramecium* to *Didinium* (i.e., *Didinium* crossmap *Paramecium*). We keep the color to be consistent with our paper so it's easier to interpret the following results.

We found that extended CCM can indeed capture well the time delay in the

deterministic system. However, we noticed a subtle difference between time delay approach and time shift experiment in our data. For the simulated data, the time delay in Ye's paper is precisely implemented into the equation and the results of extended CCM is reasonably expected. However, for the real-world data, the time delay has to be realized by time shift of the data sample as we did in our paper. This is why our Figure 3c shows a different pattern with the Figure 1 presented in Ye's paper. Furthermore, the treatment of time shift in real-world data could be somehow problematic because the time shift is artificial and the interpretation will be a bit subjective.

Here we show the results of extended CCM and hare/lynx and wolf/moose data.

The results from other two pairs of predator and prey data shows a clearly different pattern with what is shown in *Didinium* and *Paramecium* data. The difference in results could be due to different modes of interactions between distinct predator and prey, but we think this could be a sign of inconsistency when interpreting extended CCM results.

Furthermore, we applied the extended CCM to LV model. We generated a time series that covers 21 population cycles of predator (blue) and prey (red) as shown below.

We would expect the time shift involved in extended CCM will results in a periodic

change of causal interaction between predator and prey. Instead, we found the CCM could behave oppositely in the led or lag condition and the crossmap skills (Pearson's correlation) was almost indifferntiable (note that the correlation values are approaching 1 in the crossmap of both species).

In summary, we think that extended CCM can work very well in the theoretical conditions. However, in the real-world data, the test of CCM using time shift can be inconsistent because such time shift is artificially done in the time series and is not inherited in the data. Nevertheless, the paper does improve CCM to detect bidirectional causal interactions in the deterministic system.

We have cited this important paper and added a brief discussion regarding extended CCM. However, we think the comparisons of causality methods is done primarily with original version of CCM in our paper; these results given here will be mainly for addressing Reviewer's concern. The newly added section is as follows:

"It is noteworthy that the effect of temporal shift on the CCM is relevant to the extended CCM to detect time-delayed causal interactions (Ye 2015). The extended CCM has been shown to capture bidirectional causal interactions in the deterministic system. However, in the real-world data, the time-delayed causal interaction has to be achieved by arbitrary temporal shift of time series data, and the interpretation of such results is still of concern, as demonstrated in our Figure 3c."

3-4: The application to a deterministic discrete-time competitive relationship (equations 7) show stronger effect of one variable, X, on the other, Y (Figure 2b), which is also in

agreement with Sugihara (2012), in which stronger effect is shown by faster convergence of predicting for X than for Y. The authors' approach works equally well on a discrete-time predator-prey system to which stochasticity is added (Figure 2a).

Response 3-4: In accordance with response to the Reviewer 2, the causal decomposition follows the covariance of cause and effect as proposed by Galilei, hence the method is less vulnerable to the constraint of time precedence principle and applicable to both deterministic and stochastic system once the key mode is uncovered by ensemble EMD.

3-5: The authors compare causal decomposition with three other methods of inferring causality; data length, downsampling (making the signal smaller), and temporal shift. The comparisons are very interesting. However, more explanation should be given concerning the meaning of some of the results of the three other methods. It should be noted that Ye et al. (2015, Scientific Reports) extends CCM to time-delayed causal interactions, which may be relevant to the temporal shift comparisons shown in Figure 3c.

Response 3-5: Yes, it's relevant and in accordance with the response 3-3. We examined the results section regarding Figure 3 and revised it accordingly. The description for Fig. 3c is indeed incomplete so we have added more explanation for Fig 3c. We also expand the caption for Fig. 3 and other figures in accordance with concerns by all Reviewers.

3-6: Figure 4 shows relative causal strengths for the LV model and two well-known time series. The results agree with the general view that the top-down effect of the predator is dominant. The authors might want to apply their approach to the Rosenzweig-MacArthur predator-prey model, which is much more realistic than the LV.

Response 3-6: This is very helpful comment. The LV model assumes insatiable predators, and Rosenzweig and MacArthur (RM) model (1969) limits on gut size and time available for hunting dictate that the predators' kill rate will approach an upper bound as the density of prey increases.

We adopted the RM model as follows:

$$\frac{dx}{dt} = x \left(1 - \frac{x}{k} \right) - \frac{myx}{1+x}$$

$$\frac{dy}{dt} = -cy + \frac{mxy}{1+x}$$

where x and y denoted the prey and the predator, respectively ($k = 2$, $m = 3$, and $c = 1$, were used in this study).

We generated simulated time series up to 200 time units as shown below.

Next, we extracted the data between 100 and 150 time units where the model becomes sustained and stabilized.

Using the ensemble EMD parameter $r = 0.5$, we uncovered the key mode of causal inferences between predator (blue) and preys (red) as shown below:

The key mode (IMF 4) is consistent with Figure 4 presented in the paper and shows a weak causal interaction from predator to prey. We do note that RM model is sensitive to change of parameters and less robust than LV model. So we decided to keep the presentation of LV model in the paper. The comparisons of theoretical models and their parameters may be worth to investigate in the future papers.

3-7: Figure 5. Again the text and figure caption are very brief and don't explain the meaning of the Granger and MIMC results well enough. More interpretation would be helpful here.

Response 3-7: We revised the manuscript to give more precise descriptions of results in Figure 5 (as well as Figure 3), and expanded the figure captions to make them more self-explanatory.

3-8: The causal decomposition method appears to work well at determining relative causal influences for pairs of variables, both in model systems and in time series data from real ecological systems. I believe this work is novel and is a methodology that will be of broad interest, as it has applicability in many areas of science. I would like to see a bit more discussion of the authors' note near the end of the manuscript that "this novel approach will assist with revealing causal interactions in complex network". They should outline how the approach, which is only demonstrated for pairs of variables here, might

be extended to complex systems that might have a number of variables.

Response 3-8: Again, we thank the Reviewer for encouraging and helpful comments on our paper. In accordance with response to Reviewer 2 (response **2-1**), we think the use of multi-variate EMD (Rehman 2009 and Zhang 2017) could facilitate to uncover causal interaction in complex networks, such as functional brain networks measured by EEG or functional MRI. Though the validity of multivariate EMD in separating IMFs needs to be studied further (such as shown in Zhang 2017) before implementing into causal decomposition approach.

3-9: Minor:

Line 26. Change 'scale' to 'scales' and 'from the target' to 'from that of the target'

Line 56. Change 'the total' to 'the totality'

Line 71. Change 'blended with' to 'blending'

Line 172. Change 'operating at distinct time scales' to 'operating at a distinct time scale'

Line 317. Change 'Fig. 5 showed' to 'Fig. 5 shows'

Response 3-9: Many thanks for pointing out these grammatical errors. We have updated the manuscript accordingly.

REVIEWERS' COMMENTS:

Reviewer #1 (Remarks to the Author):

The authors have provided satisfactory answers to my questions and they have properly addressed all the issues that I have raised.

Having said this, I think that the authors misunderstood my remark about restructuring the manuscript I was not suggesting the insertion of a lengthy introduction to EMD. Rather, I was suggesting that the current state of the art about EMD and EEMD should be incorporated in the Introduction, in order to highlight the novelty of the proposed method. The inserted theoretical paragraphs (lines 109-131) is now too long. Although this is not a critical point, I suggest pruning it down and inserting at least a mention to EMD in the Introduction.

Reviewer #2 (Remarks to the Author):

Dear authors

thanks for this submitted version.

I think that your methodology is correct, while I am less convinced of its intrinsic superiority and generalizability.

Also the comparison with other methods is not exactly fair. Granger causality has been implemented for nonlinear relationships and for relationships between circular or semiperiodic variables (see <https://www.sciencedirect.com/science/article/pii/S0375960109005829>).

So I think that the paper would gain if you substituted some of the unverifiable claims of superiority (the fact that you don't share your code definitely contributes to this lack of verifiability), with some more and balanced applications.

Reviewer #3 (Remarks to the Author):

The authors study causality in both real ecological time series and mathematical models in which there is bidirectional causality, in predator-prey or competitive relationships. Given some oscillatory component of the interaction, the authors use instantaneous phase coherence as a measure of a causal relationship between two variables. In order to determine phase coherence, they decompose the time series of the two variables by means of ensemble empirical mode decomposition (EEMD). This methodology decomposes the time series into orthogonal oscillatory components, or intrinsic mode functions (IMFs). A Hilbert transform is applied to calculate the instantaneous phase for each IMF. The coherence of the phase difference between the corresponding IMFs of the two time series can then be calculated. Then the influence of one time series, the source, on the other, the target, can be determined by removing an IMF from the target and recombining the time series without that IMF. If removal of an IMF that is causally related to the target, followed by recombination, results in phase coherence that is lower, that is an indication of causal influence of the source on the target. The authors' approach, causal decomposition, determines the relative causality between the two variables (whether causality is stronger in one direction than the other) rather than absolute values of causality, so 50-50 is the default when no causality is found.

Comments

I reviewed an earlier version of this manuscript. I think that the authors' movement of the

Methods section closer to the front of the manuscript is a good idea, as the novelty of the authors' approach for inferring causality is the strong point of the paper. The methodology here using decomposition and recomposition of time series, and combination of the ensemble EMD and Hilbert transform to analyze phase coherence is a new and important addition to other methods (prediction, convergent cross mapping (CCM)). The basis of instantaneous phase dependency for inferring causality differs from other approaches. I agree with the authors that their approach is fundamentally new and greatly extends the ability to analyze causality.

The authors have given careful attention to the review comments. A substantial number of relevant references have been added to this version. For example, the authors also now acknowledge extensions of the CCM method (Ye et al. 2015) in which the CCM has been extended to detect time-delayed causal interactions. The authors now more clearly note some of the limitations of the causal decomposition, in particular that it represents statistical causality and that, so far, the method applies to pairwise interactions. I think that the revisions address the problems identified in the review.

Despite these limitations, and that the causality determined between the pairs is relative, not absolute, the method has great potential in its present form, and is perhaps extendable to multivariate systems.

Minor suggestions;

Line 194. 'occur' might be better than 'present' here.

Line 206. Although I think there is little chance for confusion, 'omega' used here as a weighting factor is also used as frequency (Line 128).

Line 222. Change 'recalculate' to 'recalculating'

Line 246. Change 'system' to 'systems'

Line 305. Change 'issue' to 'issues'

Line 306. This reads better with a comma after 'shift' and 'the' before 'predictive'

Line 327. The phrase 'absent of causality' seems awkward here. Does it mean 'of the absence of causality' ?

Line 349. Change 'is' to 'was', as it refers to previous understanding.

Line 372. This sentence reads better if 'causality largely' is changed to 'causality that is largely'

Line 381. Change 'with combing' to 'by combining'

Line 386. Change 'in the existing' to 'from the existing'

Line 388. Change 'emphasize' to 'emphasizes'

Line 398. Change 'bypass' to 'bypasses'

Line 403. Change 'with' to 'from'

Line 421. Change 'for' to 'from' and 'Though' to 'Although'

Line 430. Change 'anticipated' to 'anticipate'

Line 603. 'decreased' may be better than 'declined' here.

Line 617. I think 'that' should be removed.

Line 625. Change 'is' to 'it'

Line 642. The phrase 'absent of causality' seems awkward here. Does it mean 'of the absence of causality' ?

Reviewer #1 (Remarks to the Author):

The authors have provided satisfactory answers to my questions and they have properly addressed all the issues that I have raised.

Having said this, I think that the authors misunderstood my remark about restructuring the manuscript I was not suggesting the insertion of a lengthy introduction to EMD. Rather, I was suggesting that the current state of the art about EMD and EEMD should be incorporated in the Introduction, in order to highlight the novelty of the proposed method. The inserted theoretical paragraphs (lines 109-131) is now too long. Although this is not a critical point, I suggest pruning it down and inserting at least a mention to EMD in the Introduction.

Response: We thank the Reviewer for approving our revision. After reviewing our revised manuscript and previous Reviewers' comments, we felt that line 109-131 might still be necessary for the following reason. Despite the fact that EMD has numerous citations, EMD involves some critical ideas that are not well perceived by the audience from the biomedical field. The section will increase the readability and generalizability of our paper with a detailed introduction to the EMD method.

We do agree that it's necessary to insert a mention to EMD in the introduction. Also in Nature journals the method has to be presented after the discussion. In response to Reviewer's suggestion and given the format requirement by Nature Communications, we now added necessary introduction of the method (including EMD) in the last section of Introduction to improve the flow of manuscript. So the readers will not get confused when reading the Results directly after the Introduction.

Reviewer #2 (Remarks to the Author):

Dear authors

thanks for this submitted version.

I think that your methodology is correct, while I am less convinced of its intrinsic superiority and generalizability.

Also the comparison with other methods is not exactly fair. Granger causality has been implemented for nonlinear relationships and for relationships between circular or semiperiodic variables (see

<https://www.sciencedirect.com/science/article/pii/S0375960109005829>).

Response: We thank the Reviewer for approving the validity of our work. While the comparison with other methods is necessary for demonstrating the important difference between methods, we have provided extensive examples both in the paper and in the rebuttal letter to illustrate the validity and generalizability of the causal decomposition method. Therefore, we think it's reasonable to follow Nature Communication's policy to publish the rebuttal letter to improve the transparency and validity of this work.

So I think that the paper would gain if you substituted some of the unverifiable claims of superiority (the fact that you don't share your code definitely contributes to this lack of verifiability), with some more and balanced applications.

Response: We understand the concern about the verifiability. We will definitely share our code upon the publication of the method. We just don't want to prematurely release the code before the completion of the peer reviews as this paper is a methodological work. In addition, the ensemble EMD has been already publicly available at <http://rcada.ncu.edu.tw/research1.htm>. In fact, we use the same code to develop causal decomposition and conduct computational analysis, and anyone can literally use that code to replicate the causal decomposition approach presented in this paper.

Overall, in accordance with the policy of Nature Communications, we indicate the Code Availability in this revised manuscript and submit the full set of codes (not only the EMD code but also the code for causal decomposition) to the public domain of GitHub upon the submission of this revision.

Reviewer #3 (Remarks to the Author):

The authors study causality in both real ecological time series and mathematical models in which there is bidirectional causality, in predator-prey or competitive relationships. Given some oscillatory component of the interaction, the authors use instantaneous phase coherence as a measure of a causal relationship between two variables. In order to determine phase coherence, they decompose the time series of the two variables by means of ensemble empirical mode decomposition (EEMD). This methodology decomposes the time series into orthogonal oscillatory components, or intrinsic mode functions (IMFs). A Hilbert transform is applied to calculate the instantaneous phase for each IMF. The coherence of the phase difference between the corresponding IMFs of the two time series can then be calculated. Then the influence of one time series, the source, on the other, the target, can be determined by removing an IMF from the target and decomposing the time series without that IMF. If removal of an IMF that is causally related to the target, followed by decomposition, results in phase coherence that is lower, that is an indication of causal influence of the source on the target. The authors' approach, causal decomposition, determines the relative causality between the two variables (whether causality is stronger in one direction than the other) rather than absolute values of causality, so 50-50 is the default when no causality is found.

Comments

I reviewed an earlier version of this manuscript. I think that the authors' movement of the Methods section closer to the front of the manuscript is a good idea, as the novelty of the authors' approach for inferring causality is the strong point of the paper. The methodology here using decomposition and recomposition of time series, and combination of the ensemble EMD and Hilbert transform to analyze phase coherence is a new and important addition to other methods (prediction, convergent cross mapping (CCM)). The basis of instantaneous phase dependency for inferring causality differs from other approaches. I agree with the authors that their approach is fundamentally new and greatly extends the ability to analyze causality.

The authors have given careful attention to the review comments. A substantial number of relevant references have been added to this version. For example, the authors also

now acknowledge extensions of the CCM method (Ye et al. 2015) in which the CCM has been extended to detect time-delayed causal interactions. The authors now more clearly note some of the limitations of the causal decomposition, in particular that it represents statistical causality and that, so far, the method applies to pairwise interactions. I think that the revisions address the problems identified in the review.

Despite these limitations, and that the causality determined between the pairs is relative, not absolute, the method has great potential in its present form, and is perhaps extendable to multivariate systems.

Response: We thank the Reviewer for thoughtful and comprehensive comments, and the appreciation of this work. We do agree that the presentation of method closer to the front of manuscript is better. However, because of format requirement by Nature Communications that Method section has to be presented after the Discussion. We need to make necessary revisions to append a concise introduction of causal decomposition in the last section of Introduction, so to maintain the flow and readability of the paper.

Minor suggestions;

Line 194. 'occur' might be better than 'present' here.

Line 206. Although I think there is little chance for confusion, 'omega' used here as a weighting factor is also used as frequency (Line 128).

Line 222. Change 'recalculate' to 'recalculating'

Line 246. Change 'system' to 'systems'

Line 305. Change 'issue' to 'issues'

Line 306. This reads better with a comma after 'shift' and 'the' before 'predictive'

Line 327. The phrase 'absent of causality' seems awkward here. Does it mean 'of the absence of causality' ?

Line 349. Change 'is' to 'was', as it refers to previous understanding.

Line 372. This sentence reads better if 'causality largely' is changed to 'causality that is largely'

Line 381. Change 'with combing' to 'by combining'

Line 386. Change 'in the existing' to 'from the existing'

Line 388. Change 'emphasize' to 'emphasizes'

Line 398. Change 'bypass' to 'bypasses'

Line 403. Change 'with' to 'from'

Line 421. Change 'for' to 'from' and 'Though' to 'Although'

Line 430. Change 'anticipated' to 'anticipate'

Line 603. 'decreased' may be better than 'declined' here.

Line 617. I think 'that' should be removed.

Line 625. Change 'is' to 'it'

Line 642. The phrase 'absent of causality' seems awkward here. Does it mean 'of the absence of causality' ?

Response: We gratefully appreciate your comprehensive review of our manuscript. We have specifically revised our manuscript in accordance with each of these comments. In particular, for line 206, we changed the letter from omega to capital W for differentiating weight function (W) from frequency (omega).